# Translation of Mutant Repetitive Genomic Sequences in *Hirsutella sinensis* and Changes in the Secondary Structures and Functional Specifications of the Encoded Proteins

**DOI:** 10.3390/ijms252011178

**Published:** 2024-10-17

**Authors:** Xiu-Zhang Li, Yu-Ling Li, Ya-Nan Wang, Jia-Shi Zhu

**Affiliations:** 1State Key Laboratory of Plateau Ecology and Agriculture, Qinghai Academy of Animal Science and Veterinary, Qinghai University, Xining 810016, China or xiuzhang@qhu.edu.cn (X.-Z.L.); or 1991990033@qhu.edu.cn (Y.-L.L.); 2State Key Laboratory of Quality Ensurance and Sustainable Use of Dao-di Herbs, National Resource Center for Chinese Materia Medica, China Academy of Chinese Medical Sciences, Beijing 100700, China; wangyn@nrc.ac.cn; 3Institute of Biopharmaceutical and Health Engineering, Shenzhen International Graduate School, Tsinghua University, Shenzhen 518055, China

**Keywords:** repetitive copies of authentic genes in multiple genomic loci, repeat-induced point mutations (RIP), insertion/deletion, transversion and transition point mutations, transcriptional silencing of the 5.8S gene, genomically independent AT-biased genotypes of *Ophiocordyceps sinensis*

## Abstract

Multiple repetitive sequences of authentic genes commonly exist in fungal genomes. AT-biased genotypes of *Ophiocordyceps sinensis* have been hypothesized as repetitive pseudogenes in the genome of *Hirsutella sinensis* (GC-biased Genotype #1 of *O. sinensis*) and are generated through repeat-induced point mutation (RIP), which is charactered by cytosine-to-thymine and guanine-to-adenine transitions, concurrent epigenetic methylation, and dysfunctionality. This multilocus study examined repetitive sequences in the *H. sinensis* genome and transcriptome using a bioinformatic approach and revealed that 8.2% of the authentic genes had repetitive copies, including various allelic insertions/deletions, transversions, and transitions. The transcripts for the repetitive sequences, regardless of the decreases, increases, or bidirectional changes in the AT content, were identified in the *H. sinensis* transcriptome, resulting in changes in the secondary protein structure and functional specification. Multiple repetitive internal transcribed spacer (ITS) copies containing multiple insertion/deletion and transversion alleles in the genome of *H. sinensis* were GC-biased and were theoretically not generated through RIP mutagenesis. The repetitive ITS copies were genetically and phylogenetically distinct from the AT-biased *O. sinensis* genotypes that possess multiple transition alleles. The sequences of Genotypes #2–17 of *O. sinensis*, both GC- and AT-biased, were absent from the *H. sinensis* genome, belong to the interindividual fungi, and differentially occur in different compartments of the natural *Cordyceps sinensis* insect–fungi complex, which contains >90 fungal species from >37 genera. Metatranscriptomic analyses of natural *C. sinensis* revealed the transcriptional silencing of 5.8S genes in all *C. sinensis*-colonizing fungi in natural settings, including *H. sinensis* and other genotypes of *O. sinensis*. Thus, AT-biased genotypes of *O. sinensis* might have evolved through advanced evolutionary mechanisms, not through RIP mutagenesis, in parallel with GC-biased Genotype #1 of *H. sinensis* from a common genetic ancestor over the long course of evolution.

## 1. Introduction

Natural *Cordyceps sinensis* is one of most valued therapeutic agents in traditional Chinese medicine and has a rich history of clinical use in health maintenance, disease amelioration, post-disease and post-surgery recovery, and antiaging therapy [1,2,3]. The Chinese Pharmacopeia defines natural *C. sinensis* as an insect–fungal complex that includes the remains of a *Hepialidae* moth larva, including an intact body wall of fair thickness with numerous bristles, an intact larval intestine, head tissues, and fragments of other larval tissues [4,5], and >90 fungal species from >37 genera [6,7,8,9,10,11,12,13,14], with the differential co-occurrence of some of the 17 genotypes of *Ophiocordyceps sinensis* in different compartments of natural *C. sinensis* [4,5,15,16,17,18,19,20,21,22,23,24,25,26,27,28], i.e., the natural *C. sinensis* insect–fungi complex ≠ the entomopathogenic fungus *O. sinensis*. However, since the 1840s, the Latin name *C. sinensis* has been indiscriminately used for both the natural insect–fungi complex and the teleomorph/holomorph of the *O. sinensis* fungus [4,29,30]. The fungus was renamed *Ophiocordyceps sinensis* using *Hirsutella sinensis* strain EFCC7287 (GC-biased Genotype #1 of *O. sinensis*) as the nomenclature reference, whereas the indiscriminately used name of the natural and cultivated insect–fungi complexes has remained unchanged [4,31]. Since the improper implementation of the “One Fungus = One Name” nomenclature rule in *O. sinensis* research [32,33,34], the name *O. sinensis* has been used to refer to multiple teleomorphic and anamorphic fungi that have distinct genomes and has been indiscriminately applied to the natural insect–fungi complex [4,29]. The confusion surrounding the use of the Latin name continues and has even spread from the scientific community to the public media and mass markets, causing a significant decrease in the wholesale and retail prices of natural *C. sinensis*. Because a consensus regarding a Latin name for the natural insect–fungal product has not been reached by taxonomists and because multiple genotypes of *O. sinensis* with undetermined taxonomic positions are referred to by the same Latin name [4,5,29,33], we follow the GenBank taxonomic annotations and temporarily refer to the 17 genotypes of the fungus/fungi as *O. sinensis*, including Genotype #1 of *H. sinensis*, and we continue customarily to refer to the natural and cultivated insect–fungi complexes as *C. sinensis*; however, this practice only partially ameliorates the academic confusion arising from the indiscriminate use of Latin names and will likely be replaced by the exclusive use of distinct Latin names in the future. Mycologists Zhang et al. [35,36] also recognized the embarrassing situation of the indiscriminate use of Latin names and suggested the use of the non-Latin term “Chinese cordyceps” for the natural insect–fungi complex; however, this proposal has not been generally accepted by scholars from other scientific disciplines because governmental regulations worldwide require every natural medicinal product to have a unique, exclusive Latin name.

Although Wei et al. [20] hypothesized that the GC-biased Genotype #1 of *H. sinensis* is the sole anamorph of the solely teleomorphic *O. sinensis*, numerous studies have identified the differential co-occurrence of 17 mutant genotypes of *O. sinensis* in natural *C. sinensis*, which are indiscriminately referred to by the same Latin name [4,5,16,19,22,23,24,26,27,28,29,30,37,38]. Stensrud et al. [19] reported the existence of three phylogenetic clades of mutant *O. sinensis* sequences: Group A (GC-biased Genotype #1) and Groups B–C (AT-biased Genotypes #4–5), the variations of which “far exceed what is normally observed in fungi … even at higher taxonomic levels (genera and family)”. These authors suggested that, on the basis of the variations in the 5.8S gene sequences, Groups A–C of *O. sinensis* “share a common ancestor” and originated from “accelerated nrDNA evolution”. Zhang et al. [39] conducted a GenBank data mining study involving 397 ITS sequences annotated as belonging to *C. sinensis* or *O. sinensis* and confirmed the existence of three phylogenetic clades of mutant *O. sinensis* sequences: Clade A (GC-biased genotypes) and Clades B–C (AT-biased Genotypes #4–5). In contrast to the homogenous GC-biased Group A sequences containing only Genotype #1 reported by Stensrud et al. [19], Clade A reported by Zhang et al. [39] was quite divergent, containing the GC-biased sequences of Genotype #1 *H. sinensis* and several other GC-biased mutant sequences that were phylogenetically distant from Genotype #1 of *H. sinensis*.

The molecular heterogeneity of *O. sinensis* and the multicellular heterokaryotic structure of natural *C. sinensis* challenge the sole anamorph hypothesis for *H. sinensis* (GC-biased Genotype #1 of *O. sinensis*) [4,19,22,40]. In response to this challenge, Li et al. [41] proposed the “ITS pseudogene” hypothesis for all AT-biased genotypes “in a single genome” of the GC-biased Genotype #1 of *H. sinensis* on the basis of (1) the identification of heterogeneous ITS sequences of both GC-biased Genotype #1 and AT-biased Genotype #5 of *O. sinensis* in eight out of fifteen clones following 25 days of liquid incubation of *C. sinensis* mono-ascospores, and (2) the identification of the 5.8S gene cDNA of Genotype #1 but not of Genotype #5 in a cDNA library of strain 1220, one of eight genetically heterogeneous, ascosporic clones. Under the “ITS pseudogene” hypothesis, the simultaneous identification of homogenous Genotype #1 in seven other clones derived from the same mono-ascospores was apparently ignored. Li et al. [41] overgeneralized the “ITS pseudogene” hypothesis to all AT-biased genotypes (#4–6 and #15–17) of *O. sinensis* on the basis of insufficient evidence while attempting to justify AT-biased genotypes as pseudogenes in the single genome of GC-biased *H. sinensis* under the sole anamorph hypothesis for *H. sinensis* [20]. Li et al. [42] further postulated that all AT-biased *O. sinensis* genotypes “could have emerged either before or after” a new GC-biased “ITS haplotype was generated” through “repeat-induced point mutations (RIP)” that cause C-to-T (cytosine-to-thymine) and G-to-A (guanine-to-adenine) transitions and co-occur as AT-biased, nonfunctional repetitive internal transcribed spacer (ITS) copies in “a single genome” of GC-biased *H. sinensis*.

Kinjo and Zang [16] and Mao et al. [27] reported that *O. sinensis* genotypes share the same *H. sinensis*-like morphological and growth characteristics. Xiao et al. [22] concluded that the variable genotypic sequences likely belong to independent *O. sinensis* fungi on the basis of the differential coexistence of multiple *O. sinensis* mutants in the stroma and caterpillar body of natural *C. sinensis* samples at different maturation stages. In the debate regarding “cryptic species” [19] or independent fungi of *O. sinensis* [22], Zhang et al. [39] disproved the former and supported the latter on the basis of the results of Kimura 2-parameter analysis. Wei et al. [21] reported the sole GC-biased Genotype #1 teleomorph of *O. sinensis* in natural *C. sinensis* but the phylogenetically distinct, AT-biased Genotype #4 teleomorph of *O. sinensis* in cultivated *C. sinensis*, contradicting the anamorphic inoculants of GC-biased *H. sinensis* strains that were used in the product-orientated industrial cultivation project.

Li et al. [43] reported the differential occurrence, alternative splicing, and differential transcription and silencing of the mating-type genes of the *MAT1-1* and *MAT1-2* idiomorphs and pheromone receptor genes in the genome and transcriptome of *H. sinensis*. The gene expression patterns invalidated the “self-fertilization” hypothesis under homothallism and pseudohomothallism [40,44] and instead suggested the self-sterility of *H. sinensis* and the requirement of mating partners to accomplish sexual reproduction using physiological heterothallism or hybridization [43]. Figure 3 of [40] illustrates the multicellular heterokaryotic structure of *C. sinensis* hyphae and ascospores, which include multiple mononucleated, binucleated, trinucleated, and tetranucleated cells, likely containing different genetic materials, as predicted by Zhang and Zhang [45].

Owing to the unavailability of pure cultures of AT-biased *O. sinensis* genotypes for taxonomic determination, which prevents genomic, transcriptomic, multigene, and multilocus analyses of hypothetical “pseudogenes”, the present bioinformatic study examined the repetitive sequences of 104 of 1271 authentic genes at multiple loci of the genome and transcriptome of GC-biased Genotype #1 of *H. sinensis* and determined whether the *H. sinensis* genome is the target of RIP mutagenesis and concomitant epigenetic methylation, resulting in multiple repetitive sequences as genomic pseudogenes with a loss of the transcription and translation functions. The analysis of the repetitive genomic sequences was extended to examine the “ITS pseudogene” hypothesis that AT-biased genotypes formed after a new GC-biased “ITS haplotype was generated” in “a single genome” of *H. sinensis* through “RIP” mutagenesis. We discuss the genetic and phylogenetic characteristics of multiple repetitive ITS copies in the genome of Genotype #1 of *O. sinensis*, the genomic independence of AT-biased genotypes, and whether sufficient evidence exists to conclude that the 5.8S genes of AT-biased genotypes are permanently nonfunctional pseudogenes.

## 2. Results

### 2.1. Multilocus Analysis of Repetitive Sequences of Authentic Genes in the H. sinensis Genome and Transcriptome

To examine the hypothesis that repetitive sequences are present in a single genome of GC-biased *H. sinensis* and are generated through “RIP” mutagenesis, it is necessary to analyze the repetitive sequences of genes in multiple loci of the *H. sinensis* genome and transcriptome. Unfortunately, GenBank does not provide annotations for authentic genes in the genome assemblies ANOV00000000, JAAVMX000000000, LKHE00000000, LWBQ00000000, and NGJJ00000000 of the *H. sinensis* strains Co18, IOZ07, 1229, ZJB12195, and CC1406-203, respectively [44,46,47,48,49]. Cross-referencing on the basis of gene annotations for one (JAACLJ010000002) of the thirteen genome contigs of the *O. camponoti-floridani* strain EC05 [50], which contains 5,126,525 bp for 1271 authentic genes, the *H. sinensis* genes and their repetitive genomic copies were then positioned and annotated in the genome assemblies of five *H. sinensis* strains [44,46,47,48,49], and their transcription was analyzed in the mRNA transcriptome of the *H. sinensis* strain L0106 [51].

#### 2.1.1. Outline of the Repetitive Copies of Authentic Genes in the *H. sinensis* Genome

Among the 1271 authentic *H. sinensis* genes shown in Table 1, 1167 had no repetitive genomic copies; 37 had repetitive copies in only one *H. sinensis* genome; and 67 had multiple repetitive copies in the genomes of *H. sinensis* strains 1229, CC1406-203, Co18, IOZ07, and ZJB12195 [44,46,47,48,49]. The transcripts of the authentic genes and their repetitive copies were found in the mRNA transcriptome of the *H. sinensis* strain L0106 [51].

The repetitive copies of 36 of the 37 genes in only one *H. sinensis* genome were homologous to the authentic genes. However, one gene (32562→34036 within LWBQ01000158) encoding a PNGase family protein had a single repetitive copy (414324←415851 within LWBQ01000021) in the genome of the *H. sinensis* strain ZJB12195 [48]. This repetitive sequence showed 96.5% similarity to the authentic gene with a 53 nt segment insertion (in blue in Appendix A) but no transition or transversion point mutations. The AT content decreased slightly from 35.1% in the authentic gene to 34.9% in the repetitive copy.

The authentic gene was normally transcribed (transcript 610→1996 within GCQL01000547 in the transcriptome of the *H. sinensis* strain L0106 [51], encoding a PNGase family protein (accession #EQL03505 containing 462 aa residues)), according to the GenBank protein annotation for the genome of the *H. sinensis* strain Co18 [44]. The authentic gene contains an 87 nt intron (33112→33198 within LWBQ01000158), which is normally spliced (Appendix A). The 53 nt insertion in the repetitive sequence (415626←415574 of LWBQ01000021) was not transcribed.

Compared with the sequence EQL03505 of PNGase family proteins of the *H. sinensis* strain Co18 [44], both the authentic gene and its repetitive copy in strain ZJB12195 [48] are 180 aa shorter at the N-terminus (1→180 of EQL03505). EQL03505 showed 73.8–82.8% similarity to the PNGase family proteins of 48 other fungi.

The 67 authentic genes shown in Table 1 had multiple repetitive copies in the genome assemblies of *H. sinensis* strains 1229, CC1406-203, Co18, IOZ07, and ZJB12195 [44,46,47,48,49]. Compared with the sequences of the authentic genes, the repetitive copies of six of the sixty-seven genes presented essentially no changes (↑ or ↓ within ±1%) in the AT content in all five *H. sinensis* genome assemblies. The repetitive sequences of the remaining 61 authentic genes presented various changes (decreases, increases, or bidirectional changes) in the AT content.

#### 2.1.2. Slight Decreases in the AT Content of Repetitive Copies in the Genomes of Five *H. sinensis* Strains 

As shown in Table 1, eleven and seven authentic genes (the query sequences) had multiple repetitive genomic copies (the subject sequences) with slight decreases (**↓** ≤ 5%) or large decreases (**↓** > 5%) in the AT content, respectively, in comparison with the sequences of authentic genes for the triose-phosphate transporter (Table 2 in Section 2.1.2) and heavy metal tolerance protein precursor (Appendix A in Section 2.1.3), which are examples of the two groups of authentic genes.

Table 2 shows that repetitive copies (124108←124983 within LKHE01003657; 751126→752000 within NGJJ01000759; 27035→27910 within ANOV01000747; 9078524←9079399 within JAAVMX010000005; and 268003→268878 within LWBQ01000030) of the authentic gene for the triose-phosphate transporter (71694←72581 within LKHE01003487; 202613→203498 within NGJJ01001434; 5923→6447 and 5261→5919 within ANOV01001461; 1710624→1711511 within JAAVMX010000003; and 1755599→1756257 and 1756277→1756590 within LWBQ01000001, respectively) had a 2.1% reduction in the AT content [44,46,47,48,49], which was attributed to a greater number of T-to-C and A-to-G transitions (70) and T-to-G and A-to-C transversions (58) than of postulated RIP mutagenesis-related C-to-T and G-to-A transitions (62) and G-to-T and C-to-A transversions (21). Overall, the number (128) of A or T to G or C mutations was much greater than the number (83) of G or C to A or T mutations in the repetitive copies in the five *H. sinensis* genomes, resulting in 68.1% sequence similarity and a reduction in AT content from 44.2% in the authentic genes to 42.1% in the repetitive copies.

As shown in Table 2, the authentic genes for the triose-phosphate transporter (the genome segments within ANOV01001461, JAAVMX010000003, LKHE01003487, LWBQ01000001, and NGJJ01001434) were 98.4–100% homologous and were normally transcribed [44,46,47,48,49]. The transcript (451→1271 within GCQL01008460) in the transcriptome of the *H. sinensis* strain L0106 [51] was 100% homologous to the gene for the protein (accession #EQL01658 containing 440 aa residues [44]), with query coverages of 79–99%. EQL01658 in GenBank was annotated as the triose-phosphate transporter for the genome of the *H. sinensis* strain Co18 and showed 61.6–80.8% similarity to the triose-phosphate transporters of 29 other microbes (*Colletotrichum* sp., *Coniochaeta ligniaria*, *Diplogelasinospora grovesii*, *Leotiomycetes* sp., *Phialemonium atrogriseum*, *Ophiocordyceps* sp., *Rhexocercosporidium* sp., and *Thozetella* sp.).

The repetitive sequences (the genome segments within ANOV01000747, LKHE01003657, JAAVMX010000005, LWBQ01000030, and NGJJ01000759; *cf.* Table 2) were also normally transcribed. The transcript (1←821 within GCQL01008629 of the *H. sinensis* strain L0106) for the triose-phosphate transporter (accession #EQL02567 containing 330 aa residues [44]) was 94.3–94.5% similar to the repetitive copy sequences with a 48 nt deletion, with a query coverage of 99%. Both EQL01658 and EQL02567, which are encoded by the authentic genes and their repetitive copies, respectively, were annotated as the triose-phosphate transporter; however, these two protein sequences were only 65.4% similar to each other (Figure 1), with a query coverage of 89%.

EQL02567, encoded by the *H. sinensis* genomic repetitive copies, exhibited (1) 80.3% similarity to the sequence XP_044716728 for the eamA-like transporter family domain-containing protein of *Hirsutella rhossiliensis* strain HR02, (2) 61.3–67.4% similarity to 20 protein sequences of the solute carrier family 35 member E3 proteins of many microbes (*Colletotrichum* sp., *Coniochaeta* sp., *Diaporthaceae* sp., *Hymenoscyphus varicosporioides*, *Lasiosphaeria hispida*, *Pyricularia oryzae*, *Tolypocladium ophioglossoides*, and *Tolypocladium paradoxum*), and (3) 59.7–68.4% similarity to 26 protein sequences of the triose-phosphate transporters of many other microbes (*Amylocarpus encephaloides*, *Colletotrichum* sp., *Coniochaeta ligniaria*, *Diplogelasinospora grovesii*, *Phialemonium atrogriseum*, *Rhexocercosporidium* sp., *Thozetella* sp., *Xylariaceae* sp., and *Zalerion maritima*).

As shown in red and blue in both the upper and lower segments in Figure 1, there are repeat segments (187→276 and 287→440) within the authentic protein sequence EQL01658 (440 aa) encoded by the authentic gene of *H. sinensis* strain L0106. The repeat segments aligned to the same sequence segment (164→253) within EQL02567 (330 aa) encoded by the genomic repetitive sequence with sequence identities of 50% and 68% or positives of 77% and 82%, respectively, The N-terminal (1→36) and middle segments (277→286) of EQL01658 did not match the protein sequence of EQL02567.

Figure 2 shows ExPASy ProtScale plots for the α-helices (Panel A), β-sheets (Panel B), β-turns (Panel C) and coils (Panel D) of the triose-phosphate transporter [43,52,53]. Each panel contains two plots in pairs for EQL01658 and EQL02567. The repeated protein segments are shown with open boxes in blue in all the EQL01658 plots in Figure 2, corresponding to the segment sequences in red and blue shown in Figure 1.

The paired plots for EQL01658 and EQL02567 in all panels of Figure 2 have completely different plotting topologies and waveforms, indicating substantially variable secondary structures of the proteins encoded by the *H. sinensis* authentic gene and its genomic repetitive sequence, although both proteins were annotated as the triose-phosphate transporter [52,53,54,55,56].

Among the eleven authentic genes outlined in Table 1 that had multiple repetitive copies with slightly reduced AT content (**↓** ≤ 5%), the repetitive copies of four authentic genes presented mutagenetic patterns similar to those of triose-phosphate transporters, with a combination of more RIP-related C-to-T transitions than T-to-C transitions but more A-to-G transitions than RIP-related G-to-A transitions, or vice versa. The repetitive copies of six other authentic genes had consistently more T-to-C and A-to-G transitions than the postulated RIP-related C-to-T and G-to-A transitions did, in addition to numerous transversion, insertion, and deletion point mutations.

#### 2.1.3. Large Decreases in the AT Content of Repetitive Genomic Copies

Appendix A shows the authentic genes for the heavy metal tolerance protein precursor (2772←5944 within LKHE01001285; 195005→198178 within NGJJ01001482; 2485←5657 within ANOV01008830; 1309808←1312980 within JAAVMX010000003; and 1338987←1342159 within LWBQ01000001), which had multiple repetitive copies with >5% decreases in the AT content [44,46,47,48,49].

The repetitive copies of the authentic gene could be divided into two groups.

The first group of repetitive copies (3819←4233 within LKHE01002445; 150283→150697 within NGJJ01001243; 3175→3589 within ANOV01010958; 833901←834315 within JAAVMX010000002; and 468456←468870 within LWBQ01000010) in the genomes of the *H. sinensis* strains 1229, CC1406-203, Co18, IOZ07, and ZJB12195, respectively, were 65.4% similar to the authentic gene. These repetitive copies had many more T-to-C and A-to-G transitions (26) and A-to-C and T-to-G transversions (27) than did the postulated RIP-related C-to-T and G-to-A transitions (14) and C-to-A and G-to-T transversions (14). Overall, the total number (28) of G or C to A or T point mutations was much fewer than the number (53) of A or T to G or C mutations in the repetitive copies, resulting in 6.2% reduced AT content, from 39.5% in the authentic gene to 33.3% in the repetitive copies.The second group of repetitive copies (128317←128839 and 128885←129014 within LKHE01001747; 461774←461903 and 461949←462471 within NGJJ01001310; 4224←4746 and 4792←4921 within ANOV01005573; 85112←85634 and 85680←85809 within JAAVMX010000012; and 141513→142035 and 142081→142210 within LWBQ01000044) of the authentic gene in the five *H. sinensis* genome assemblies presented 67.4% similarity to the authentic genes. There were more T-to-C and A-to-G transitions (34) and A-to-C and T-to-G transversions (34) than postulated RIP-related C-to-T and G-to-A transitions (24) and C-to-A and G-to-T transversions (21). Overall, this group of repetitive copies had a total of 45 G or C to A or T point mutations, which was much fewer than the number (68) of A or T to G or C point mutations in the repetitive copies, resulting in a 3.7% reduced AT content, from 38.3% in the authentic gene to 34.6% in the repetitive copies.

Additionally, as shown in Appendix A, the authentic genes (the genome segments within ANOV01008830, JAAVMX010000003, LKHE01001285, LWBQ01000001, and NGJJ01001482) were transcribed normally. The fragmented transcripts (GCQL01012210, GCQL01006458, and GCQL01000615 of the *H. sinensis* strain L0106) were 99.9–100% identical to the authentic gene for the heavy metal tolerance protein precursor (accession #EQK98634 containing 1029 amino acids [44]), with query coverages of 12–32%. EQK98634 showed 74.0–82.3% similarity to the heavy metal tolerance protein precursor of 19 other microbes (*Drechmeria coniospora*, *Fusarium* sp., *Hapsidospora chrysogenum*, *Metarhizium* sp., *Moelleriella libera*, *Ophiocordyceps* sp., *Pochonia chlamydosporia*, *Purpureocillium* sp., and *Tolypocladium* sp.).

The first group of repetitive copies (the genome segments within ANOV01010958, JAAVMX010000002, LKHE01002445, LWBQ01000010, and NGJJ01001243) was 65.4% similar to the authentic gene and was transcribed (Appendix A). The transcript GCQL01017466 was 100% homologous to the DNA sequences of repetitive copies, with a query coverage of 100%; however, it presumably encoded a different protein, the transporter ATM1 (accession #EQK98286, which contains 598 amino acids [44]), according to the GenBank protein annotation for the genome of the *H. sinensis* strain Co18. EQK98286 was 84.8–89.4% similar to 59 protein sequences of the mitochondrial iron–sulfur cluster transporter ATM1 of different microbes (*Akanthomyces lecanii*, *Beauveria* sp., *Colletotrichum* sp., *Conoideocrella luteorostrata*, *Cordyceps* sp., *Dactylonectria macrodidyma*, *Fusarium* sp., *Ilyonectria robusta*, *Lecanicillium* sp., *Mariannaea* sp., *Metarhizium* sp., *Neonectria ditissima*, *Ophiocordyceps* sp., *Paramyrothecium foliicola*, *Purpureocillium* sp., *Tolypocladium* sp., and *Trichoderma* sp.).

The second group of repetitive copies (the genome segments within ANOV01005573, JAAVMX010000012, LKHE01001747, LWBQ01000044, and NGJJ01001310) was 67.4% similar to the authentic genes and was also transcribed (Appendix A). The fragmented transcripts (GCQL01013313 and GCQL01016959) were 99.8–100% homologous to the repetitive copies with query coverages of 38–61%; however, the fragmented transcripts presumably encoded a different protein, the ABC transporter (accession #EQK99315, containing 844 amino acids [44]), according to the GenBank protein annotation for the genome of the *H. sinensis* strain Co18. EQK99315 was 50.2–89.0% similar to 22 other ABC transporter protein sequences of many different microbes (*Aspergillus* sp., *Beauveria bassiana*, *Blastomyces* sp., *Chaetomium strumarium*, *Histoplasma* sp., *Penicillium* sp., *Purpureocillium* sp., and *Ophiocordyceps* sp.).

The EQK98634 sequence encoded by the authentic gene showed only 38% and 37% similarity to the EQK99315 and EQK98286 sequences, respectively, encoded by the repetitive genomic sequences, representing proteins with altered specificities of heavy metal clustering and transport, namely, the heavy metal tolerance protein precursor, transporter ATM1, and the ABC transporter. The protein sequences (EQK99315 and EQK98286) encoded by the repetitive sequences showed only 32.8% similarity. ExPASy ProtScale plotting analyses demonstrated very different plotting topologies and waveforms (Appendix A), indicating distinct secondary structures of the different proteins.

Like the case of the mutagenetic pattern of the authentic gene encoding the heavy metal tolerance protein precursor, the repetitive copies of six other genes whose AT content decreased substantially (↓ > 5%) (*cf.* Table 1) were associated with consistently greater numbers of T-to-C and A-to-G transitions than postulated RIP-related C-to-T and G-to-A transitions, in combination with numerous transversion, insertion, and deletion point mutations.

#### 2.1.4. Slight Increases in the AT Content of Repetitive Genomic Copies

The authentic genes for the maltose permease and lipase/serine esterase are representatives of the 28 authentic genes that had multiple repetitive copies with slight increases (↑ ≤ 5%; n = 17; Appendix A) and large increases (↑ > 5%; n = 11; Table 3) in the AT content, respectively.

Appendix A shows the authentic genes for the maltose permease in the five *H. sinensis* genome assemblies (47693→49623 within LKHE01002657; 77988→79917 within NGJJ01000203; 1725←3655 within ANOV01009266; 4614206→4616136 within JAAVMX010000006; and 66747→68677 within LWBQ01000157 of *H. sinensis* strains 1229, CC1406-203, Co18, IOZ07, and ZJB12195, respectively) [44,46,47,48,49]. The repetitive sequences (6824→7385 within LKHE01000336; 782871←782913 and 1076399←1076960 within NGJJ01001217; 4433←4994 and 5254←5290 within ANOV01000509; 12448464→12449025 and 12779815→12779864 within JAAVMX010000005; and 292175→292217 and 570642→571203 within LWBQ01000003; *cf.* Appendix A) of the authentic gene showed 69.6–70.4% sequence similarity to the authentic gene sequence. These repetitive sequences had more T-to-C and A-to-G transitions (35–37) than did the postulated RIP-related C-to-T and G-to-A transitions (30–31), as well as more C-to-A and G-to-T transversions (28) than A-to-C and T-to-G transversions (10–11). Overall, there were more G or C to A or T point mutations (58–59) than A or T to G or C point mutations (46–47) in the repetitive copies, resulting in slight increases (↑ ≤5%) in the AT content, from 36.8–37.9% in the authentic genes to 38.5–39.9% in the repetitive copies.

Appendix A also shows that the authentic genes (the genome segments within ANOV01009266, JAAVMX010000006, LKHE01002657, LWBQ01000157, and NGJJ01000203) were transcribed normally. Transcript GCQL01010488 was 99.8–100% homologous to the authentic genes for the maltose permease (accession #EQK98556 according to the GenBank protein annotation for the genome of the *H. sinensis* strain Co18 [44], with a query coverage of 80%. EQK98556 showed 74.6–84.9% similarity to the maltose permease or α-glucoside permease of 17 other microbes (*Drechmeria coniospora*, *Fusarium* sp., *Metarhizium* sp., *Ophiocordyceps* sp., *Tolypocladium* sp., and *Trichoderma citrinoviride*).

The repetitive copies (the genome segments within ANOV01000509, JAAVMX010000005, LKHE01000336, LWBQ01000003, and NGJJ01001217; *cf.* Appendix A) were transcribed, and two overlapping transcripts (GCQL01015773 and GCQL01019033) were 98.7% identical, with a query coverage of 100%. GCQL01015773 was 100% homologous to the repetitive copies, and GCQL01019033 was 95.6% similar to the repetitive copies, with a 25 nt deletion. These two transcripts encoded a different protein (MRT, a fungal-unique gene encoding a member of the raffinose family of oligosaccharide transporters, 152→338 of EQL02995, containing 550 amino acids, according to the GenBank protein annotation for the genome of the *H. sinensis* strain Co18 [44]; however, the protein encoded by GCQL01019033 had a deletion of eight amino acids (Appendix A). The N-terminal segment (1→100) and the C-terminal segment (612→636) of EQK98556 did not match the N-terminal segment (1→15) or the C-terminal segment (525→550) of EQL02995. EQL02995 showed 73.7–77.2% similarity to the MRTs of eight other microbes (*Metarhizium* sp., *Moelleriella libera*, and *Ophiocordyceps* sp.).

Appendix A shows ExPASy ProtScale plots for the α-helices (Panel A), β-sheets (Panel B), β-turns (Panel C) and coils (Panel D) of the maltose permease and the protein in the raffinose family of oligosaccharide transporters. Each panel contains two ProtScale plots in pairs for EQK98556 and EQL02995, which have distinct plotting topologies and waveforms, indicating substantially variable conformations of the proteins encoded by the authentic gene and its genomic repetitive sequence, although both proteins function in oligosaccharide inter/intracellular transport.

In addition to the authentic gene for the maltose permease shown in Appendix A, 16 other authentic *H. sinensis* genes were found, as outlined in Table 1, with slight increases in the AT content. Seven of them exhibited a mutagenetic pattern with contradictory transitions (T-to-C and A-to-G vs. RIP-related C-to-T and G-to-A) and transversions (T-to-G and A-to-C vs. C-to-A and G-to-T) in combination with various numbers of insertion and deletion point mutations, similar to the mutagenetic pattern for the maltose permease. Nine other genes had consistently more RIP-related C-to-T and G-to-A transitions than T-to-C and A-to-G transitions; more C-to-A and G-to-T transversions than T-to-G and A-to-C transversions; and many insertion and deletion point mutations.

#### 2.1.5. Large Increases in the AT Content of Repetitive Genomic Copies

Table 3 shows the authentic genes for the lipase/serine esterase (22031→22947 within LKHE01002746; 235828→236719 within NGJJ01000647; 9368→10284 within ANOV01000049; 10692268←10693159 within JAAVMX010000005; and 203574←204490 within LWBQ01000026) in the five *H. sinensis* genomes [44,46,47,48,49]. The repetitive sequences (39965→40763 and 40848→40982 within LKHE01001863; 1146814→1147612 and 1147697→1147831 within NGJJ01001576; 6319←6453 and 6538←7335 within ANOV01000484; 22221540←22222338 within JAAVMX010000001; and 513365→514163 and 514248→514382 within LWBQ01000013) were 67.0–68.5% similar to the authentic genes in the five *H. sinensis* genome assemblies. There were more postulated RIP-related C-to-T and G-to-A transitions (63) than T-to-C and A-to-G transitions (30) and more C-to-A and G-to-T transversions (41) than A-to-C and T-to-G transversions (20) in the repetitive copies. Overall, there were more G or C to A or T point mutations (104) in the repetitive sequences than A or T to G or C point mutations (50), resulting in large increases (↑ >5%) in the AT content from 34.9–35.0% in the authentic genes to 41.5–41.6% in the repetitive copies.

**Table 3 ijms-25-11178-t003:** Authentic *H. sinensis* genes for lipase/serine esterase (query sequence) and repetitive genomic copies with major increases (>5%) in the AT content.

The Subject Sequence (Repetitive Copy)		*vs*. The Query Sequence (235828→236719 of NGJJ01000647 of the Authentic Gene for the Lipase/Serine Esterase)		Mutation in the Subject Sequence Comparing with the Query Sequence						Transcript in mRNA Transcriptome GCQL00000000			
*H. sinensis* Strain	Genomic Fragment (Sequence Range)	Similarity	Change in AT content	Transitions		Transversions		Total Point Mutations		Transcriptomic Fragment	Similarity	Query Coverage	Translated Protein Sequence
				C-to-T and G-to-A	T-to-C and A-to-G	C-to-A and G-to-T	A-to-C and T-to-G	G or C to A or T	A or T to G or C				
1229	LKHE01002746 (22031→22947)	100%	No ∆ (34.5%)							GCQL01000519 (1663→2554) or GCQL01015314 (2847→3477)	100% 100%	100% 70%	EQL04141 (325→621) EQL04141 (270→479)
	LKHE01001863 (39965→40763; 40848→40982)	68.5%	↑ 35.0% to 41.6%	63	30	41	20	104	50	GCQL01000532 (5013→6030)	100%	100%	EQL03018 (265→587)
CC1406-203	NGJJ01000647 (235828→236719)	100%								GCQL01000519 (1663→2554) or GCQL01015314 (2847→3477)	100% 100%	100% 70%	EQL04141 (325→621) EQL04141 (270→479)
	NGJJ01001576 (1146814→1147612; 1147697→1147831)	68.5%	↑ 35.0% to 41.6%	63	30	41	20	104	50	GCQL01000532 (5013→6030)	100%	100%	EQL03018 (265→587)
Co18	ANOV01000049 (9368→10284)	99.8%	No ∆ (34.5%)							GCQL01000519 (1638→2554) or GCQL01015314 (2847→3477)	100% 100%	100% 68%	EQL04141 (325→621) EQL04141 (270→479)
	ANOV01000484 (6319←6453; 6538←7335)	67.0%	↑ 34.9% to 41.5%	63	30	41	20	104	50	GCQL01000532 (5013←6030)	99.7%	100%	EQL03018 (265→587)
IOZ07	JAAVMX010000005 (10692268←10693159)	100%	No ∆ (34.5%)							GCQL01000519 (1663→2554) or GCQL01015314 (2847→3477)	100% 100%	100% 70%	EQL04141 (325→621) EQL04141 (270→479)
	JAAVMX010000001 (22221540←22222338)	67.1%	↑ 34.9% to 41.6%	63	30	41	20	104	50	GCQL01000532 (5013←5811)	100%	100%	EQL03018 (265→587)
ZJB12195	LWBQ01000026 (203574←204490)	100%	No ∆ (34.5%)							GCQL01000519 (1638←2554) or GCQL01015314 (2847→3477)	100% 100%	100% 68%	EQL04141 (325→621) EQL04141 (270→479)
	LWBQ01000013 (513365→514163; 514248→514382)	68.5%	↑ 35.0% to 41.6%	63	30	41	20	104	50	GCQL01000532 (5013→5811)	100%	100%	EQL03018 (265→587)

Note: “→” and “←” indicate the sequence directions of the sense and antisense strands, respectively. “**↑**” indicates an increase in the AT content.

Table 3 also shows that the highly homologous (99.8–100%) authentic genes (the segments within ANOV01000049, JAAVMX010000005, LKHE01002746, LWBQ01000026, and NGJJ01000647) were transcribed. Two transcripts (GCQL01000519 and GCQL01015314) were 100% homologous to the authentic gene and encoded partially overlapping protein sequences of lipase/serine esterase (accession #EQL04141 (1016 aa)) [44], according to the protein annotation for the genome of the *H. sinensis* strain Co18, with query coverages of 68–100%. EQL04141 showed 64.3% similarity to the putative serine esterase of *Hirsutella rhossiliensis* and 40.4–44.5% similarity to the lipase/serine esterases of 19 other microbes (*Beauveria bassiana*, *Dactylonectria estremocensis*, *Fusarium albosuccineum*, *Ilyonectria* sp., *Mariannaea* sp., *Metarhizium* sp., *Ophiocordyceps* sp., *Paramyrothecium foliicola*, *Pochonia chlamydosporia*, *Purpureocillium* sp., *Thelonectria olida*, and *Tolypocladium* sp.).

The repetitive copies (the segments within ANOV01000484, JAAVMX010000001, LKHE01001863, LWBQ01000013, and NGJJ01001576; *cf.* Table 3) were 67.0–68.5% similar to the authentic gene and were normally transcribed. The transcript (GCQL01000532) was 99.7–100% homologous to the repetitive copies, with a query coverage of 100%; however, it encoded the hypothetical proteins OCS_01262 and G6O67_000668 (accession #EQL03018 and KAF4513393), according to the protein annotations for the genomes of the *H. sinensis* strains Co18 and IOZ07, respectively. EQL03018 (1116 aa [44]) and KAF4513393 (1132 aa [49]) were only 46.7% and 47.2%, respectively, similar to EQL04141 [44], which is encoded by an authentic gene, with a query coverage of 94%. EQL03018 and KAF4513393 encoded by the repetitive genomic sequences showed high homology (98.5%) to protein sequences identified from different *H. sinensis* strains and were 61.5–83.6% similar to many protein sequences for (1) **the lipase/serine esterase** of *Pochonia chlamydosporia* and *Purpureocillium* sp.; (2) **the serine esterase** of *Hirsutella rhossiliensis*, *Ilyonectria robusta*, *Metarhizium acridum*, *Thelonectria olida*, and *Trichoderma breve*; and (3) **the lipase-like proteins** of *Fusarium* sp., *Hapsidospora chrysogenum*, *Metarhizium anisopliae*, *Tolypocladium* sp., and *Trichoderma lentiforme*. Thus, the protein sequence EQL04141 (encoded by the authentic *H. sinensis* gene) and the protein sequences EQL03018 and KAF4513393 (encoded by the repetitive copies) with approximately 47% similarity might represent lipid and ester hydrolases with different substrate specificities, i.e., in theory, hydrolyzing water-insoluble long-chain triacylglycerols by lipases or water-soluble short acyl chain esters by esterases [57].

The authentic protein sequence EQL04141 and the protein sequences of the genomic repetitive sequences KAF4513393 and EQL03018 were compared and are shown in Figure 3. The EQL04141 sequence appears to be divided into two parts, (1) 53→837 and (2) 807→1014, with an overlap of 20 aa residues. These two segments aligned to 8→833 and 877→1101 of KAF4513393 with 40 residue insertions in between KAF4513393 and 8→817 and 861→1085 of EQL03018 with 44 residue insertions in between the EQL03018 sequence.

Figure 4 shows ExPASy ProtScale plots for the α-helices (Panel A), β-sheets (Panel B), β-turns (Panel C) and coils (Panel D). Each panel contains two plots in pairs for EQL04141 encoded by the authentic gene of lipase/serine esterase and KAF4513393 encoded by the repetitive sequence for other esterases or lipases, which showed distinct plotting topologies and waveforms, indicating significant variations in the KAF4513393 protein conformation with altered catalytic specificities in lipid metabolism compared with those of authentic lipase/serine esterase. The plots for KAF4513393 essentially have identical topologies and waveforms to those for EQL03018; the sequences of both are highly homologous, as shown in Figure 3.

The repetitive copies of 10 other *H. sinensis* genes with a largely increased AT content (*cf.* Table 1) consistently presented more RIP-related C-to-T and G-to-A transitions than T-to-C and A-to-G transitions, in addition to transversion, insertion, and deletion point mutations.

#### 2.1.6. Bidirectional Changes in the AT Content of Repetitive Genomic Copies

Table 4 shows the authentic genes for the β-lactamase/transpeptidase-like proteins (19330→20932 within LKHE01000540; 70191→71142 within NGJJ01001580; 2857←4459 within ANOV01000528; 238583←240185 within JAAVMX010000002; and 168259←169259 within LWBQ01000052) among the 15 authentic genes listed in Table 1 that had multiple repetitive copies with bidirectional changes in the AT content in the five *H. sinensis* genome assemblies [44,46,47,48,49]. The multiple repetitive copies could be divided into three groups, two of which presented a reduced AT content but slightly different patterns of point mutations, and the other group presented an increased AT content.

The first group of repetitive copies with slight decreases (↓ ≤ 5%) in the AT content (54230←54311 and 54418←55104 within LKHE01001740 and 883241→883927 and 884034→884115 within JAAVMX010000009; *cf.* Table 4) had an equal number (68) of RIP-related C-to-T and G-to-A transitions and opposite T-to-C and A-to-G transitions but more A-to-C and T-to-G transversions (40) than C-to-A and G-to-T transversions (26). Overall, the repetitive copies contained more A or T to G or C point mutations (108) than G or C to A or T point mutations (94), resulting in a sequence similarity of 67.9% to the authentic gene and a 5% decrease in the AT content, from 38.0% in the authentic genes to 33.0% in the repetitive copies.

The second group of repetitive copies with a slightly reduced AT content (697372→698058 and 698165→698246 within NGJJ01000083; 136→549 and 656→737 within ANOV01000033; and 1407095←1407176 and 1407283←1407696 within LWBQ01000002; *cf.* Table 4) showed 67.9–70.4% similarity to the authentic gene and had more T-to-C and A-to-G transitions (110 and 27) than postulated RIP-related C-to-T and G-to-A transitions (85 and 16) and more A-to-C and T-to-G transversions (71 and 15) than C-to-A and G-to-T transversions (40 and 14). Overall, there were fewer total G or C to A or T point mutations (125 and 30) in the repetitive copies than A or T to G or C point mutations (181 and 44), which was attributed to 2.4–5.0% decreases in the AT content, from 35.4–38.0% in the authentic gene sequences to 32.7–33.0% in the repetitive sequences.

The third group of repetitive copies with an increased AT content is shown in Table 4 (20238←21185 within LKHE01002381; 250601→251549 within NGJJ01000732; 15299→16246 within ANOV01000652; 16799894←16800841 within JAAVMX010000003; and 210989←211936 within LWBQ01000004), which exhibited 64.2% similarity to the authentic gene. There were more postulated RIP-related C-to-T and G-to-A transitions (74) than T-to-C and A-to-G transitions (46) and more C-to-A and G-to-T transversions (48) than A-to-C and T-to-G transversions (43) in the repetitive sequences. Overall, the repetitive copies had more G or C to A or T point mutations (122) than A or T to G or C point mutations (89), resulting in 3.9–4.0% increases in the AT content, from 38.0% in the authentic genes to 41.9–42.0% in the repetitive copies.

Additionally, as shown in Table 4, the authentic genes (the genome segments within ANOV01000528, JAAVMX010000002, LKHE01000540, LWBQ01000052, and NGJJ01001580) were transcribed and encoded β-lactamase/transpeptidase-like proteins. Among the three fragmented transcripts, GCQL01006885 and GCQL01011125 were 99.6–100% homologous to the authentic gene, with query coverages of 17–43%, whereas the other transcript, GCQL01011658, was 92.1% similar (with a 58 nt deletion) to the authentic genes, with query coverages of 29–49%. The fragmented transcripts encoded nonoverlapping sequences of the β-lactamase/transpeptidase-like protein with sequence gaps (accession #EQL02970, containing 531 amino acids), according to protein annotation for the *H. sinensis* strain Co18 [44].

The repetitive copies with a reduced AT content (the genome segments within ANOV01000033, JAAVMX010000009, LKHE01001740, LWBQ01000002, and NGJJ01000083; *cf.* Table 4) were transcribed. The transcripts (GCQL01006426, GCQL01008668, and GCQL01020269) were 98.2–100% homologous to the sequences of the repetitive copies, with query coverages of 21–65% and short segment overlaps and gaps in protein sequences. The transcripts presumably encoded the hypothetical protein G6O67_008083 (accession #KAF4504658, containing 542 amino acids [49]), according to the protein annotation for the *H. sinensis* strain IOZ07. KAF4504658 was 46% similar, with 62% positives when compared to the authentic protein sequence EQL02970 of *H. sinensis* (Figure 5), and was 49.1–80.4% similar to 24 GenBank protein sequences for the β-lactamase/transpeptidase-like protein of many different microbes (*Dactylonectria* sp., *Fusarium* sp., *Ilyonectria* sp., *Mariannaea* sp., *Microthyrium microscopicum*, *Ophiocordyceps* sp., *Thelonectria* olida, and *Trichoderma* sp.).

The repetitive copies with an increased AT content (the genome segments within ANOV01000652, JAAVMX010000003, LKHE01002381, LWBQ01000004, and NGJJ01000732; *cf.* Table 4) were transcribed. The GCQL01012138 transcript was 94.6% similar (with a 51 nt deletion) to the repetitive sequences and encoded a β-lactamase/transpeptidase-like protein (accession #EQL02706, containing 558 aa, according to the protein annotation for the *H. sinensis* strain Co18 [44]), with 100% query coverage. EQL02706 was 50% similar, with 66% positives to the authentic protein sequence EQL02970 of *H. sinensis* (Figure 5).

The protein sequences of KAF4504658 and EQL02706 [44,49], which are encoded by repetitive sequences, were 49.3% similar to each other. These sequences were 46.2% and 50.0% similar and 62% and 66% positive, respectively, to EQL02970, which is encoded by the authentic gene encoding the β-lactamase/transpeptidase-like protein (Figure 5) [44]. The ExPASy ProtScale plots of EQL02970, KAF4504658, and EQL02706 in all panels of Figure 6 have completely different plotting topologies and waveforms, indicating significant variations in the KAF4504658 and EQL02706 protein conformations with altered catalytic specificities compared with those of authentic β-lactamase-transpeptidase.

The sequences of EQL02706 and KAF4504658 were (1) 47.9–80.4% similar to those of **the β-lactamase/transpeptidase-like proteins** of various microbes (*Akanthomyces lecanii*, *Dactylonectria* sp., *Fusarium* sp., *Hirsutella rhossiliensis*, *Ilyonectria* sp., *Mariannaea* sp., *Metarhizium* sp., *Microthyrium microscopicum*, *Thelonectria* sp., and *Trichoderma* sp.) and (2) 49.3–63.3% similar to those of **the penicillin-binding proteins** of various microbes (*Drechmeria coniospora*, *Fusarium* sp., *Pochonia chlamydosporia*, *Purpureocillium* sp., and *Trichoderma arundinaceum*).

### 2.2. Multiple GC-Biased Repetitive ITS Copies in the H. sinensis Genome

Fungal genomes contain multiple repetitive copies of the nrDNA ITS1-5.8S-ITS2 sequences. Three genome assemblies (ANOV00000000, LKHE00000000, and LWBQ00000000) of the *H. sinensis* strains Co18, 1229, and ZJB12195, respectively [44,46,47], were uploaded to GenBank prior to 2020. Only single copies of the ITS1-5.8S-ITS2 sequences could be identified in each of these genomes (the segments within ANOV01021709, LKHE01000582, and LWBQ01000008), probably because the repetitive ITS sequences were discarded during the assembly of the genome shotgun reads, which indicated that Li et al. [41] might not have been able to obtain real and non-abridged genomic data for the analysis of repetitive ITS copies before manuscript submission as the basis for backup of their “ITS pseudogene” hypotheses.

However, multiple repetitive ITS copies were identified in two additional genome assemblies, JAAVMX000000000 and NGJJ00000000, for the *H. sinensis* strains IOZ07 and CC1406-203, respectively [48,49], which were available in GenBank after June 2020. Seventeen repetitive ITS1-5.8S-ITS2 copies were found within JAAVMX000000000, seven repetitive ITS1-5.8S-ITS2 copies were found within NGJJ00000000 (Table 5), and two partial ITS segments within NGJJ00000000 were not included in our analysis.

Most of the repetitive ITS sequences listed in Table 5 were highly homologous (98.8–100%) to AB067721, the reference sequence of the GC-biased Genotype #1 of *H. sinensis*. Two variable repetitive copies (6233→6733 and 44729→45251) within JAAVMX010000019 presented 94.5% and 90.8% similarity to AB067721, respectively. Two other copies, 18702095→18702586 within JAAVMX010000002 and 700→1186 within JAAVMX010000018, showed similarities of 97.4% and 97.0%, respectively, to AB067721. The repetitive ITS sequences presented 80.1–89.9% similarity to the AT-biased Genotype #4–6 and #15–17 sequences of *O. sinensis* (Table 5).

### 2.3. Genetic Characteristics of the Multiple Repetitive ITS Sequences in the H. sinensis Genome

The repetitive ITS sequences within the genome assemblies JAAVMX000000000 and NGJJ00000000 for the *H. sinensis* strains IOZ07 and CC1406-203, which are listed in Table 5, and the single genomic ITS sequences within ANOV01021709, LKHE01000582, and LWBQ01000008 for the *H. sinensis* strains Co18, 1229, and ZJB12195, respectively, which are listed in Appendix A [44,46,47,48,49], were GC biased; their average GC content was 64.7 ± 0.33% (*cf.* Table 5), which was nearly identical to the GC content (64.8%) of the AB067721 Genotype #1 sequence but significantly greater than that of the AT-biased Genotypes #4–6 and #15–17, i.e., 51.1 ± 1.69% (44.8–53.1%; *p* < 0.001) [4,5]. These genetic features of the variable repetitive ITS copies were distinct from those of AT-biased *O. sinensis* genotypes, which included multiple scattered transition alleles (Figure 7).

The sequences of repetitive ITS copies, 18702095→18702586 within JAAVMX010000002 and 700→1186 within JAAVMX010000018 (*cf.* Table 5), were less variable (97.4% and 97.0% similarity to AB067721); they contained no transition alleles and predominantly included 13 and 15 insertion and deletion, and transversion alleles, with allelic ratios of 13:0 and 15:0, respectively, for insertion/deletion/transversion vs. transition (in green in Figure 7; Table 6). These genetic characteristics of the variable and less variable repetitive ITS copies indicate that they were not generated through RIP mutagenesis, which theoretically causes cytosine-to-thymine (C-to-T) and guanine-to-adenine (G-to-A) transition mutations. Other repetitive ITS copies with 98.1–99.8% similarity to AB067721 and containing only a few insertion/deletion point mutations are listed in Table 5.

### 2.4. Phylogenetic Features of Multiple Repetitive ITS Copies in the H. sinensis Genome

As shown in Table 5 and Appendix A and Figure 7, the sequences of GC-biased Genotypes #2–3 and #7–14 and AT-biased Genotypes #4–6 and #15–17 of *O. sinensis* are absent in the genome assemblies ANOV00000000, JAAVMX000000000, LKHE00000000, LWBQ00000000, and NGJJ00000000 of the *H. sinensis* strains Co18, IOZ07, 1229, ZJB12195, and CC1406–203, respectively, and instead belong to the genomes of independent *O. sinensis* fungi [4,5,19,21,22,23,25,39,44,46,47,48,49].

The genomic repetitive ITS copies were phylogenetically clustered into the GC-biased Genotype #1 clade, which is shown in blue alongside the Bayesian majority rule consensus tree in Figure 8, and were phylogenetically distant from the AT-biased clades containing Genotypes #4–6 and #15–17 of *O. sinensis*, shown in red alongside the tree. This phylogenetic feature is similar to what has previously been demonstrated in phylogenetic trees that were conferred using different algorithms [4,5,16,19,21,30,39]. Figure 8 includes the two variable genomic repetitive ITS copies with low similarity (94.5% and 90.8%) and the two other repetitive ITS copies with relatively low similarity (97.4% and 97.0%), which are shown in Table 5 and Appendix A. The GC-biased repetitive ITS copies are individually scattered within the Genotype #1 clade. The two variable repetitive ITS sequences, 6233→6733 (94.5% similarity) and 44729→45251 (90.8% similarity), within the JAAVMX010000019 genome assembly (*cf.* Table 5 and Appendix A) presented greater phylogenetic distances (horizontal lines) from other GC-biased repetitive ITS copies within the GC-biased clades. GC-biased Genotypes #2–3 and #7–14 of *O. sinensis* presented greater phylogenetic distances from the sequences of GC-biased Genotype #1 and the repetitive ITS copies. These results indicate that genetic variations in the GC-biased repetitive ITS sequences shown in blue in Figure 7 had a significant impact on the Bayesian phylogenetic clustering analysis.

### 2.5. The Repetitive Genomic ITS Sequences Related to Multiple GC-Biased Genotypes of O. sinensis

In addition to the phylogenetic analysis of multiple genotypes of *O. sinensis* shown in Figure 8, the sequences of GC-biased Genotypes #1–3 and #7–12 were further analyzed (Appendix A) and compared with the genomic repetitive ITS copies. GC-biased Genotypes #13–14 were characterized by large DNA segment reciprocal substitutions and recombination of genetic material between the genomes of the two parental fungi, Genotype #1 of *H. sinensis* and an AB067719-type fungus [5], and they were not included in the analysis in Appendix A. Figure 8 shows that GC-biased Genotypes #2–3 and #8–14 formed several GC-biased branches outside the Genotype #1 clade with great phylogenetic distances (horizontal lines), and they were phylogenetically and genetically distinct from the genomic repetitive ITS copies of Genotype #1 of *H. sinensis* (Appendix A, Appendix A).

Genotype #7 was unique among the GC-biased genotypes and was placed within the Genotype #1 clade with great phylogenetic distance (horizontal line) in the Bayesian tree in Figure 8. The Genotype #7 sequence (AJ488254) was originally identified in the stroma of a *C. sinensis* sample (H1023) collected from Qinghai Province, China, and uploaded to GenBank by Chen et al. [25]. The authors identified the GC-biased Genotype #1 ITS sequence (AJ488255) from the tail of the caterpillar body of the same sample, leading to questions regarding whether the sequences of AJ488254 and AJ488255 coexisted in a single genome of GC-biased Genotype #1 *H. sinensis* as two repetitive ITS copies or belonged to the genomes of independent *O. sinensis* fungi co-occurring in different compartments of the same *C. sinensis* sample. The study conducted by Chen et al. [25] did not involve the purification of *O. sinensis* strain(s) within the same natural sample. Thus, there is no evidence to date for determining whether the sequence (AJ488254) of Genotype #7 is indeed present in the genome of Genotype #1 of *H. sinensis* as one of the repetitive ITS copies, although Genotype #7 was placed within the Genotype #1 phylogenetic clade shown in Figure 8 and was phylogenetically distant from GC-biased Genotypes #2–3 and #8–14, which were placed outside the Genotype #1 clade in the Bayesian tree shown in Figure 8.

Notably, GC-biased Genotypes #1 and #2 were simultaneously detected in the stromata of immature, maturing, and mature *C. sinensis* samples collected from Kangding County in Sichuan Province [37]. These two genotypes presented different developmental patterns in an asynchronous, disproportional manner in *C. sinensis* stromata during maturation [37], indicating the genomic independence of the two GC-biased genotypes. The Bayesian tree shown in Figure 8 confirms that the sequences of Genotypes #1 and #2 were placed in separate GC-biased phylogenetic clades.

The GC-biased repetitive ITS sequences that were variable (94.5% and 90.8% similarity) in comparison with the reference sequence AB067721 of Genotype #1 predominantly contained insertion/deletion point mutations and included less frequent transversion alleles, whereas only a few transition alleles were found (Table 6). The variable ITS repeats 6233→6733 and 44729→45251 within JAAVMX010000019 (Table 5) contained twenty-two and thirty-two insertion/deletion alleles and four and eleven transversion alleles, respectively (in blue in Figure 7; Table 6); they contained only one and five transition alleles, with allelic ratios of 26:1 and 43:5, respectively, for insertion/deletion/transversion vs. transition.

The sequences of Genotypes #2 and #7 were 94.4–95.3% similar to those of Genotype #1 (*cf.* Appendix A), containing numerous mutant alleles, which were distinct from the mutant alleles in the genomic repetitive sequences shown in Appendix A. Thus, the sequences of Genotypes #2 and #7 were unlikely to be represented as repetitive ITS copies in a single *H. sinensis* genome.

### 2.6. Transcriptional Silencing of 5.8S Genes

The genetic, genomic, and phylogenetic analyses (Table 5, Table 6 and Appendix A; Figure 7, Figure 8 and Appendix A) demonstrated the GC-biased characteristics of the multiple repetitive ITS copies in the *H*. *sinensis* genome assemblies and that all the GC- and AT-biased genotypes of *O*. *sinensis* were genome independent. Li et al. [41] identified ITS sequences for both Genotypes #1 and #5 from genomic DNA that was isolated from eight (1206, 1208, 1209, 1214, 1220, 1224, 1227, and 1228) of fifteen clones of natural *C. sinensis* multicellular heterokaryotic mono-ascospores [40]. However, Li et al. [41] reported 5.8S cDNA only for Genotype #1 and not for Genotype #5 in a cDNA library constructed from total RNA extracted from Clone 1220, one of the eight genetically heterogeneous clones. However, they did not report cDNA library construction for seven other genetically heterogeneous clones (1206, 1208, 1209, 1214, 1224, 1227, and 1228) or subsequent cDNA examinations for Genotypes #1 and #5 of *O. sinensis*.

Li et al. [41] did not detect the ITS sequences of AT-biased Genotypes #4, #6, and #15–17 in the genomic DNA isolated from any of the 15 ascosporic clones. Thus, there is insufficient evidence to conclude that the 5.8S genes of all AT-biased genotypes of *O. sinensis* are permanently nonfunctional pseudogenic components coexisting in “a single genome” with a functional copy of GC-biased Genotype #1. To respond to this academic challenge by Zhu et al. [38], Li et al. [58] provided the following logical reasoning underlying the nonfunctional pseudogene hypothesis: “If AT-biased ITS sequences are not pseudogenes, 5.8S cDNA should have been detected because rRNAs are critical and essential for protein synthesis in all living organisms”; this logical reasoning critically touches on two serious scientific issues. First, are the 5.8S genes of all the *O. sinensis* genotypes actively transcribed in *C. sinensis* ascospores? Second, is it correct to conclude that “5.8S cDNA should have been detected” by the techniques designed by Li et al. [41]?

The differential coexistence of several AT-biased genotypes has been reported in various combinations in different compartments of natural *C. sinensis* [4,5,22,26,28,37,59]. Genotypes #4 and #15 of AT-biased phylogenetic Cluster B of *O. sinensis* (*cf.* Figure 8) were reported to exist in the stroma and stromal fertile portion (densely covered with numerous ascocarps) of natural *C. sinensis* but not in the ascospores [4,5,28], which is consistent with the failure after additional specific efforts to identify Genotype #4 of AT-biased Cluster B in mono-ascospores, as reported by Li et al. [41]. Thus, there is no evidence to date that AT-biased Genotypes #4 and #15 occur in *C. sinensis* ascospores, and there is no ground for claiming that the 5.8S genes of Genotypes #4 and #15 are nonfunctional pseudogenes in the genome of GC-biased Genotype #1.

Xiang et al. [60] reported a metatranscriptomic study of natural *C. sinensis* samples (unknown maturation stage) collected from Kangding County in Sichuan Province, China. A GenBank BLAST search confirmed that the metatranscriptome GAGW00000000 did not contain any 5.8S gene transcripts of *C. sinensis*-associated fungi (including all 17 genotypes of *O. sinensis*), indicating the transcriptional silencing of the 5.8S genes of all co-colonizing fungi in natural settings. Xia et al. [61] reported another metatranscriptomic study of mature natural *C. sinensis* samples collected from Deqin County in Yunnan Province, China. Similarly, this assembled metatranscriptome assembly contained no 5.8S gene transcripts. The absence of 5.8S gene transcripts in the metatranscriptome assemblies indicates the transcriptional silencing of the 5.8S genes of multiple *O. sinensis* genotypes in natural settings [60,61].

In contrast, Li et al. [41] reported the detection of a 5.8S gene transcript after a 25-day in vitro inoculation of mono-ascospores. The author reported that the 5.8S transcript belonged to GC-biased Genotype #1 but not to AT-biased Genotype #5. This hypothetical conclusion might suggest that the 5.8S gene of Genotype #1 was non-naturally transcribed in in vitro culture experimental settings but that the 5.8S gene of Genotype #5 might maintain its natural silenced state.

The second question is whether the methods designed by Li et al. [41] are suitable for scientifically identifying 5.8S gene cDNAs of multiple *O. sinensis* genotypes [42]. Li et al. [58] retrospectively explained the study design logic for the 5.8S-F/R primer pair that was used by Li et al. [41]: “The 5.8S-F/R primers … located in the most conserved region”, indicating an experimental design with greatly diminishing the specificity of the primers for 5.8S cDNA. The primer pair 5.8S-F/R was 100% and 95% homologous to the 5.8S gene sequence of AT-biased Genotype #5. In contrast to the 95% and 85% similarity to the 5.8S gene sequence of GC-biased Genotype #1, the 5.8S-F/R primer pair may be favored for amplifying the 5.8S cDNA sequence of Genotype #5 rather than for amplifying the 5.8S cDNA sequence of Genotype #1. The 5.8S-F/R primers presented variable sequence similarities, ranging from 61.9–100% and 31.8–95.5%, respectively, compared with the 17 genotypes of *O. sinensis*, which preferentially amplified the most easily amplified cDNA sequence in the cDNA library considering the primary structure and secondary conformation of the PCR templates and consequently caused false-negative results for the other 5.8S cDNA sequences.

Li et al. [41] did not disclose the sequences of the identified 5.8S cDNA but instead reported that “Of these clones, 28 showed identical sequences (104 bp excluding the primers) to the most frequent sequence in group A (JQ900148). The remaining 14 sequences … had single substitutions at various positions”, where group A = Genotype #1 *H. sinensis*. The GenBank annotation of the 5.8S gene of JQ900148 is at 192→346, covering 155 nt, which is not a multiple of three and is not consistent with the open reading frame of the gene.

Alternatively, the 5.8S genes were annotated at the following nucleotide positions: 218→373 within AB067721 of Genotype #1 and AB067740 of Genotype #5, as well as 214→369 within AB067744 of Genotype #4. These sequences covered 156 nt and 52 codons in the three cases [30]. Excluding the 5.8S-F/R primer regions, the remaining sequence of the 5.8S gene consisted of 114 bp, with 10 additional nucleotides (mismatches) beyond the 104 bp in the 28 cDNA clones reported by Li et al. [41]. Upon adding “single substitutions at various positions” to 14 other clones, as reported by Li et al. [41], the 42 clones presumably contained either 10 or 11 mismatches (8.8–9.6%) in the 114 bp segment of the 5.8S gene cDNA, corresponding to 90.4–91.2% sequence similarity to the 5.8S gene of *H. sinensis*. Such low similarities indicated the unlikelihood of the 5.8S gene cDNA detected by Li et al. [41] truly belonging to GC-biased Genotype #1. Alternatively, the nondisclosure of the 42 detected cDNA sequences did not exclude the possibility that the detected cDNA sequences were derived from AT-biased Genotype #5 of *O. sinensis* or from non-rRNA genes.

Alternatively, the 5.8S genes were annotated at the following nucleotide positions: 218→373 within AB067721 of Genotype #1 and AB067740 of Genotype #5, as well as 214→369 within AB067744 of Genotype #4. These sequences covered 156 nt and 52 codons in the three cases [4,30]. Excluding the 5.8S-F/R primer regions, the remaining sequence of the 5.8S gene consisted of 114 bp, with 10 additional nucleotides (mismatches) beyond the 104 bp in the 28 cDNA clones reported by Li et al. [41]. Upon adding “single substitutions at various positions” to 14 other clones, as reported by Li et al. [41], the 42 clones presumably contained either 10 or 11 mismatches (8.8–9.6%) in the 114 bp segment of the 5.8S gene cDNA, corresponding to 90.4–91.2% sequence similarity to the 5.8S gene of *H. sinensis*. Such low similarities indicated the unlikelihood of the 5.8S gene cDNA detected by Li et al. [41] truly belonging to GC-biased Genotype #1. Alternatively, the nondisclosure of the 42 detected cDNA sequences did not exclude the possibility that the detected cDNA sequences were derived from AT-biased Genotype #5 of *O. sinensis* or from non-rRNA genes.

## 3. Discussion

### 3.1. Multilocus Analysis of the Repetitive Copies of Numerous Authentic Genes in the Genome of Genotype #1 H. sinensis

Selker and his colleague [62,63,64,65] reported frequent C-to-T (cytosine-to-thymine) transitions and concomitant epigenetic methylation of nearly all remaining cytosine residues in fungal 5S RNA genes associated with transcriptional silencing. Gladyshev [66] suggested that RIP “occurs specifically in haploid parental nuclei that continue to divide by mitosis in preparation for karyogamy and ensuing meiosis” and “mutates cytosines on both strands of each DNA duplex in a pairwise fashion (e.g., preferentially affecting none or both repeated sequences).” Hane et al. [67] stated that “the RIP process selectively mutated duplicated sequences in both DNA strands by inducing single-nucleotide point (SNP) mutations that converted C:G base pairs to T:A. This often led to the introduction of nonsense or missense mutations which affected the expression of these sequences.” The sequence-specific targets of RIP mutagenesis for *de novo* DNA methylation are under debate [67,68]. Sun (2021) believes that RIP mutagenesis occurs before meiosis in the sexual cycle of certain fungi and plays biological roles to balance their strengths and weaknesses and to maintain stability and advance evolution.

The repetitive sequences of multiple authentic *H. sinensis* genes were analyzed at multiple loci in the *H. sinensis* genome assemblies, namely, ANOV00000000, JAAVMX000000000, LKHE00000000, LWBQ00000000, and NGJJ00000000, and in the *H. sinensis* transcriptome assembly GCQL00000000 [44,46,47,48,49,51]. Among the 1271 authentic genes in the *H. sinensis* genome (Table 1), 104 (approximately 8.2%) had repetitive copies; 37 (approximately 2.9%) of the 104 authentic genes had repetitive genomic copies in only one *H. sinensis* genome, and 67 (approximately 5.3%) had multiple repetitive copies in all 5 *H. sinensis* genomes. Table 2, Table 3 and Table 4 and Appendix A present five representative *H. sinensis* genes with a variety of point mutations, including various postulated RIP-related C-to-T and G-to-A transitions and opposite T-to-C and A-to-G transitions, as well as C-to-A and G-to-T transversions and opposite A-to-C and T-to-G transversions, in addition to insertions/deletions. The point mutations caused a low similarity of repetitive sequences compared with the sequences of authentic genes and increases, decreases, or bidirectional changes in the AT content. Nevertheless, the long course of evolution of repetitive genes may enable *O. sinensis* to better adapt to the harsh environment (extremely cold in long winter, low oxygen partial pressure, and strong ultraviolet radiation) through mutations under natural selective pressure at high altitude on the Qinghai–Tibet Plateau, and the repetitive genomic sequences play important roles in the life cycle of natural *C. sinensis* [19,69,70,71].

In general, the repetitive copies of most authentic genes (83 of 104) at multiple *H. sinensis* genomic loci contained fewer postulated RIP-related C-to-T and G-to-A transitions than opposite T-to-C and A-to-G transitions. In contrast, the repetitive copies with large increases (↑ > 5%) in the AT content included more RIP-related C-to-T and G-to-A transitions than opposite T-to-C and A-to-G transitions in only 11 of the 104 authentic genes (*cf.* Table 3). The slight increases (↑ ≤ 5%) in the AT content in the repetitive copies of eight authentic genes (*cf.*
Appendix A) were the result of transversion mutations; however, the postulated RIP-related C-to-T and G-to-A transitions were less frequent than opposite T-to-C and A-to-G transitions were.

As shown in Table 2, Table 3 and Table 4 and Appendix A, the transcripts for the repetitive genomic copies of the authentic genes could be identified in the mRNA transcriptome GCQL00000000 for the *H. sinensis* strain L0106 [51], regardless of the increases, decreases, or bidirectional changes in the AT content and low sequence similarity of the repetitive copies compared with the authentic genes. The authentic genes and repetitive genomic copies likely encode proteins with altered functional specificities, which enhance the adaptability of natural *C. sinensis* to harsh, high-altitude environments and expand its therapeutic efficacy for disease resistance and treatment, anti-aging, and health preservation [1,2,3,19,43,52,53,71]. Thus, the genetic results of the multilocus analysis and the transcription ability of the repetitive copies further invalidated the pseudogene hypothesis and its root cause of RIP mutagenesis, as proposed by Li et al. [41,42,58]. *O. sinensis*, especially the GC-biased Genotype #1 of *H. sinensis*, may not be the mutagenetic target of RIP, which causes nonsense and missense mutations of authentic *O. sinensis* genes, concurrent epigenetic methylation attacks, and subsequent dysfunction of multiple repetitive copies of authentic genes.

### 3.2. The GC-Biased Repetitive ITS Copies in the Genome of Genotype #1 of H. sinensis Are Unrelated to RIP Mutagenesis

Figure 7 and Appendix A and Table 6 demonstrate that the variable repetitive ITS copies were characterized by multiple scattered insertion, deletion, and transversion point mutations, which were not generated through RIP mutagenesis because RIP mutagenesis theoretically causes C-to-T and G-to-A transitions. AT-biased genotypes of *O. sinensis* characterized by multiple scattered transition point mutations (Figure 7) were generated in parallel with other GC-biased genotypes through accelerated nuclear rDNA evolution from a common ancestor over the long course of evolution [19]. Thus, evidence is lacking to support the hypothesis that RIP mutagenesis induces the formation of mutant AT-biased repetitive ITS copies within the single genome of GC-biased Genotype #1 of *H. sinensis*, as postulated by Li et al. [42]. Even if AT-biased genotypes were indeed induced by RIP-like mutagenesis over the long course of evolution, there has been no evidence that AT-biased genotypes “could have emerged either before or after” a new GC-biased “ITS haplotype was generated” through RIP mutagenesis and became pseudogenic components of the single genome of GC-biased Genotype #1, as assumed by Li et al. [41,42,58].

Hane et al. [67] reported that RIP mutagenesis “is observed only in certain fungal taxa; the Pezizomycotina (filamentous Ascomycota) and some species of the Basidiomycota.” Table 5 and Table 6 and Figure 7, Figure 8 and Appendix A show that all the multiple genomic repetitive ITS copies were GC biased with no or only a few transition alleles, and were phylogenetically distinct from AT-biased genotypes, which possess numerous scattered transition alleles. Thus, it is reasonable to question whether Genotype #1 of *H. sinensis* is susceptible to RIP mutagenesis and epigenetic methylation attack.

### 3.3. Genetic and Phylogenetic Differences between the AT-Biased Genotypes of O. sinensis and GC-Biased Repetitive ITS Copies in the Genome of Genotype #1 of H. sinensis

The sequences of AT-biased Genotypes #4–6 and #15–17 of *O. sinensis* contain numerous scattered transition alleles and are genetically and phylogenetically distant from GC-biased repetitive ITS sequences containing predominant insertion, deletion, and transversion point mutations within the genome of GC-biased Genotype #1 of *H. sinensis* (*cf.* Figure 7 and Figure 8 and Table 5, Table 6 and Appendix A). The sequences of AT-biased Genotypes #4–6 and #15–17 of *O. sinensis*, as well as GC-biased Genotypes #2–3 and #7–14, are absent in the genome assemblies ANOV00000000, JAAVMX000000000, LKHE00000000, LWBQ00000000, and NGJJ00000000 of the *H. sinensis* strains Co18, IOZ07, 1229, ZJB12195, and CC1406–203, respectively (*cf.* Table 5 and Appendix A and Figure 7 and Appendix A), but instead belong to the genomes of independent *O. sinensis* fungi [4,5,22,30,38,39,44,46,47,48,49]. The non-residency of AT-biased *O. sinensis* genotypes does not support the hypothesis that the AT-biased sequences are “ITS pseudogene” components of “a single genome” of GC-biased Genotype #1 of *H. sinensis*. Stensrud et al. [19] reported that the variation in the 5.8S gene sequences of AT-biased genotypes “far exceeds what is normally observed in fungi … even at higher taxonomic levels (genera and family)”. Xiao et al. [22] concluded that the sequences of AT-biased genotypes likely belong to the genomes of independent *O. sinensis* fungi.

### 3.4. Multicellular Heterokaryons and Molecular Heterogeneity of Natural C. sinensis Hyphae and Ascospores

Bushley et al. [40] illustrated that *C. sinensis* hyphae and ascospores are multicellular heterokaryons with mononucleated, binucleated, trinucleated, and tetranucleated cellular structures. Zhang and Zhang [45] asked whether the binucleated cells (and other monokaryotic and polykaryotic cells) of *C. sinensis* hyphae and ascospores contained homogenous or heterogeneous genetic materials, suggesting that these monokaryotic and polykaryotic cells may contain heterogeneous sets of chromosomes and genomes.

Li et al. [41] identified heterogeneous ITS sequences of both GC-biased Genotype #1 and AT-biased Genotype #5 in eight of fifteen ascosporic clones (1206, 1208, 1209, 1214, 1220, 1224, 1227, and 1228) and simultaneously identified seven other clones (1207, 1218, 1219, 1221, 1222, 1225, and 1229) that contained only homogenous sequences of Genotype #1 after 25 days of in vitro inoculation with natural *C. sinensis* mono-ascospores. These culture-dependent molecular mycological findings support the morphological observations of multicellular heterokaryons discovered by their collaborators [40], although molecular mycological strategies may overlook nonculturable fungal components or conditional nonculturable components under the experimental design by Li et al. [41]. The two groups of reported ascosporic clones were likely derived from different cells of multicellular heterokaryotic mono-ascospores of natural *C. sinensis* [40], confirming the inference of Zhang and Zhang [45] that there are heterogeneous sets of chromosomes and genomes in monokaryotic and polykaryotic cells.

In addition to the fifteen clones reported by Li et al. [41], the authors did not disclose any information on other possible ascosporic clones, namely, clones 1210, 1211, 1212, 1213, 1215, 1216, 1217, 1223, and 1226, for which the clone serial numbers are missing among those of the reported two groups (*cf.* Table S1 of [41]), nor did the authors state whether these nine possible clones were simultaneously obtained in the same study along with the fifteen reported clones, and whether they contained different fungal genetic materials, although these questions should have been asked during the journal’s peer review process and should have been disclosed to the public by the authors. The Editors-in-Chief of Molecular Phylogenetics and Evolution, Drs. Derek Wildman and E.A. Zimmer, encouraged and supported discussions of the relevant scientific issues in the journal with the authors [30,38].

In contrast, Li et al. [5], from a different research group, used culture-independent techniques and reported the co-occurrence of GC-biased Genotypes #1 and #13–14 and AT-biased Genotypes #5–6 and #16 of *O. sinensis*, *P. hepiali*, and an AB067719-type fungus in *C. sinensis* ascospores. These findings from nonculture experiments confirmed the genetic heterogeneity of the multicellular heterokaryotic ascospores of natural *C. sinensis*. Zhu et al. [37], Gao et al. [26], and Li et al. [5,28] demonstrated that the *O. sinensis* genotypes underwent dynamic alterations in immature, maturing, and mature stromata; the stromal fertile portion that is densely covered with numerous ascocarps; and ascospores of natural *C. sinensis*, indicating the genomic independence of the *O. sinensis* genotypes and that many of the genotypic fungi may not be culturable and detectable in the in vitro culture-dependent experimental settings established by Li et al. [41].

### 3.5. Genomic Variations of H. sinensis Strains

By aggregating the information from molecular and genomic studies [44,46,47,48,49], different genomic DNA samples of “pure” *H. sinensis* strains were examined and published and can be summarized and grouped as follows:Group 1 *H. sinensis* strains consisted of pure, homokaryotic anamorphic *H. sinensis* strains (1229, CC1406-203, Co18, IOZ07, and ZJB12195) [44,46,47,48,49]. Total genomic DNA was isolated from these strains and subjected to genome-wide sequencing. The genomes contained no AT-biased sequences of *O. sinensis* genotypes.Group 2 *H. sinensis* strains consisted of seven clones (strains 1207, 1218, 1219, 1221, 1222, 1225, and 1229) among fifteen clones derived from 25 days of incubation of natural *C. sinensis* mono-ascospores [41]. Total genomic DNA was isolated from these strains and shown to contain the homogenous ITS sequence of Genotype #1 of *H. sinensis* but included no AT-biased sequences of *O. sinensis* genotypes.Group 3 “*H. sinensis*” strains consisted of eight other clones (strains 1206, 1208, 1209, 1214, 1220, 1224, 1227, and 1228) obtained from the 25-day incubation of *C. sinensis* mono-ascospores [41]. Total genomic DNA was isolated from these strains, which exhibited genetic heterogeneity and the coexistence of GC-biased Genotype #1 and AT-biased Genotype #5 of *O. sinensis*.

The *H. sinensis* strains of Groups 1 and 2 were similar, with each strain possessing a homogenous GC-biased Genotype #1 genome, although multiple GC-biased repetitive ITS copies were identified in the genome assemblies JAAVMX000000000 and NGJJ00000000 for the *H. sinensis* strains IOZ07 and CC1406-203, respectively (*cf.* Table 5 and Table 6 and Figure 7, Figure 8 and Appendix A) [41,44,46,47,48,49]. Although a single copy of the ITS sequence was identified in the genome assemblies ANOV00000000, LKHE00000000, and LWBQ00000000 of the *H. sinensis* strains Co18, 1229, and ZJB12195, respectively [44,46,47], the genomes of these strains may theoretically contain many repetitive ITS copies that could have been discarded during the assembly of genome shotgun reads. Conceivably, if technically possible, reassembling the genome shotgun reads may very likely identify additional repetitive ITS copies in the genomes of Group 1 strains Co18, 1229, and ZJB12195, among which strain 1229 is also a Group 2 strain that was derived from mono-ascospores by Li et al. [41]. Similarly, the genome sequencing of other Group 2 strains may very likely reveal multiple repetitive ITS copies.

The *H. sinensis* strains in Groups 2 and 3 were derived after 25 days of liquid culture of the same *C. sinensis* mono-ascospores [41]; however, they differed genetically. The Group 3 strains may have heterokaryotic mycelial structures, similar to the mycelia derived from the microcycle conidiation of *H. sinensis* ascospores initially cultured on solid medium plates for 30 days and subsequently in liquid medium for 10–53 days, as shown in Figures 3 and 4 of [59]. Li et al. [41] reported that Group 3 strains included cocultured GC-biased Genotype #1 and AT-biased Genotype #5 of *O. sinensis* fungi, which are genomically independent and coexisted in heterokaryotic mycelial cells, indicating either diploid/polyploid mycelial cells containing heterogeneous sets of chromosomes/genomes, or impure mycelia, each containing a different chromosome/genome. However, the authors did not identify or report any haploid clones containing only AT-biased Genotype #5, indicating that this AT-biased *O. sinensis* fungus may not exhibit independent culturability under in vitro experimental settings, although other studies have reported the identification of single AT-biased Genotype #4 or #5 of *O. sinensis* without coculture with GC-biased Genotype #1 in cultures of natural and cultivated *C. sinensis* under different experimental settings [16,21,27]. Unfortunately, Li et al. [41,42,58] neglected the differences in genetic material between the strains in Groups 2 and 3, which were presumably derived from the same or different cells of multicellular heterokaryotic ascospores [5,16,27,30,40,45].

In addition to the genome sequencing of the homogenous strain 1229 in both Groups 2 and 3 and the genome assembly LKHE00000000 by Li et al. [46], the genome sequencing of one of the heterogeneous Group 3 strains will provide critical genomic evidence to validate the hypothesis that AT-biased Genotype #5 of *O. sinensis* coexists as an “ITS pseudogene … in a single genome” of GC-biased Genotype #1. The authors may also consider a multilocus approach to explore pseudogenes in the heterogeneous genomes of Group 3 strains and broaden the current single locus “ITS pseudogene” hypothesis to multiple genomic loci. Unfortunately, the authors elected not to do so since 2012 or 2013 after obtaining the eight Group 3 strains or perhaps failed in such an endeavor owing to technical difficulties in the assembly of shotgun reads because heterogeneous sequences from more than one set of genomes of independent genotypic *O. sinensis* fungi evolved from the same genetic ancestor. Consequently, they elected willingly or unwillingly to forgo the opportunity to obtain and publish solid evidence to validate their “ITS pseudogene” hypothesis, which states that AT-biased Genotype #5 and GC-biased Genotype #1 physically coexist “in a single genome”. Studies from other research groups have confirmed that the sequences of all AT-biased genotypes belong to the genomes of independent fungi [4,5,19,22,26,27,30,37,38,46] and that heterokaryotic ascospores and hyphae with multiple mononucleated, binucleated, trinucleated, and tetranucleated cells contain heterogeneous sets of chromosomes and genomes [40,45].

In addition to the three groups of *H. sinensis* strains summarized above, many other *H. sinensis* strains have been studied and presumably belong to Groups 1–3 or other groups. For instance, Li et al. [72] isolated two *H. sinensis* strains, CH1 and CH2, from the intestines of healthy larvae of *Hepialus lagii* Yan. These strains exhibited *H. sinensis*-like morphological and growth characteristics but contained GC-biased Genotype #1, AT-biased Genotypes #4–5, and *P. hepiali*. The wild-type strains strongly enhanced the inoculation potency of *H. sinensis* by 15- to 39-fold in the larvae (n = 100 in each experimental group) of *Hepialus armoricanus* (*p* < 0.001). Additionally, some “pure” *H. sinensis* strains (gifts from a senior mycologist) contained *H. sinensis* and *P. hepiali* [72]. As mentioned above, Li et al. [41] did not discuss whether nine other probable ascosporic clones (1210, 1211, 1212, 1213, 1215, 1216, 1217, 1223, and 1226) simultaneously obtained in the study constituted additional fungal group(s) containing different genetic materials.

### 3.6. Transcription of the 5.8S Gene and PCR Amplicons of the 5.8S-F/R Primers

A gene transcription analysis (*cf.* Section 3.6) revealed that Li et al. [41,58] neglected the evidence of transcriptional silencing of 5.8S genes demonstrated in metatranscriptomic studies in natural settings and the evidence of dynamic and unbalanced transcriptional activation and silencing of numerous fungal genes during the continuous in vitro fermentation or liquid incubation of Genotype #1 of *H. sinensis* [51,60,61]. Changes in in vitro culture conditions and durations have significant impacts on the transcription of numerous fungal genes [51], which involves the transcriptional activation and silencing of numerous fungal genes. Owing to the conservative evolution of 5.8S genes and many other non-rDNA genes, the incomplete study design and controversial findings reported by Li et al. [41] need to be verified using other sophisticated transcription methods to address the issues of whether the cDNA identified by Li et al. [41] was truly derived from the 5.8S gene of the GC-biased Genotype #1 of *H. sinensis*, from the mutant AT-biased Genotype #5 of *O. sinensis*, from other colonized fungi, or even from non-rRNA genes, before confidently declaring that the 5.8S genes of the multiple AT-biased genotypes of *O. sinensis* are permanently nonfunctional pseudogenes under the “ITS pseudogene … in a single genome” hypothesis.

Li et al. [41] did not disclose the amplicon sequences amplified using 5.8SF/R primers or the PCR template of the cDNA library constructed via reverse transcription of total RNA derived from *H. sinensis* Clone 1220. Li et al. [58] confirmed that Li et al. [41] designed 5.8SF/R primers on the basis of the sequence “in the most conserved region” of *H. sinensis* rRNA, indicating largely reduced specificity of the primers and resulting in uncertain template sources of the PCR amplicons. Our sequence analysis revealed that the percentage of cDNA sequences detected by Li et al. [41] was only 90.4–91.2%, similar to the 5.8S gene sequence of Genotype #1 of *O. sinensis*. Such high dissimilarity rates (8.8–9.6%) may not be convincingly attributed to “mismatches during reverse transcription or PCR amplification”, as explained by Li et al. [41]. Thus, solid evidence is lacking to conclude that the cDNA sequences were truly derived from the 5.8S gene of Genotype #1 of *H. sinensis* and that the *H. sinensis* 5.8S gene was considered the functional gene, whereas the 5.8S gene of Genotype #5 of *O. sinensis* was considered the nonfunctional “ITS pseudogene”.

Prior to declaring that the 5.8S genes of all AT-biased *O. sinensis* genotypes are permanently nonfunctional pseudogenes co-occurring with a functional 5.8S gene in the genome of GC-biased Genotype #1, several other concerns should be carefully addressed.

Scientists [62,63,64,65,66,67,68,69] have repeatedly reported the epigenetic methylation of nearly all cytosine residues in 5.8S RNA genes and subsequent transcriptional silencing of GC-biased genes, which occurs only in certain fungal species. As shown in Figure 7 and Appendix A and Table 6, the multiple genomic repetitive ITS copies of GC-biased *H. sinensis* contained mainly insertion, deletion, and transversion alleles but no or only a few transition alleles. The repetitive ITS copies of Genotype #1 are genetically and phylogenetically distinct from the AT-biased genotypes of *O. sinensis* (*cf.* Figure 7 and Figure 8). Although the 5.8S gene of GC-biased Genotype #1 contains many cytosine residues, which are potentially susceptible to epigenetic methylation attack, introducing nonsense or missense mutations that subsequently cause translational silencing of 5.8S genes and the multilocus evidence, including the ITS locus presented in this paper, indicates that *H. sinensis* species may not be the target of RIP mutagenesis and epigenetic methylation attack. It is unlikely that the AT-biased genotypes of *O. sinensis* emerged immediately before or after the generation of a new *H. sinensis* genome; instead, they are likely genomically independent and exist in different *O. sinensis* fungi.Li et al. [41] did not detect the ITS sequences of Genotypes #2–4 and #6–17 of *O. sinensis* in the genomic DNA pool of mono-ascosporic cultures. Thus, it is an overgeneralization to infer that the 5.8S genes of these genotypes are nonfunctional “ITS pseudogene” components of the genome of Genotype #1 *H. sinensis*.The sequences of Genotypes #2–17 are absent in the genome assemblies ANOV00000000, JAAVMX000000000, LKHE00000000, LWBQ00000000, and NGJJ00000000 of the *H. sinensis* strains Co18, IOZ07, 1229, ZJB12195, and CC1406-2031229, respectively [44,46,47,48,49], indicating the genomic independence of multiple *O. sinensis* genotypes.The culture-dependent approach used by Li et al. [41] might have overlooked some fungal species that are nonculturable or difficult to culture under the in vitro experimental settings of the study. The culture-dependent strategy might have a significant impact on the transcription of many fungal genes, with many genes being switched on or off nonnaturally, as evidenced by the nonlinear reduction in the total number of transcriptomic unigenes and increases in the average length but decreases in the GC content of transcripts during 3–9 days of liquid fermentation of the *H. sinensis* strain L0106 [51]. A much greater impact on differential transcriptional activation and silencing of many genes of GC-biased Genotype #1 and AT-biased Genotype #5 of *O. sinensis* may be expected after the prolonged 25-day liquid incubation period adopted by Li et al. [41].Three distinct types of secondary steric conformations of the 5.8S rRNA were predicted for Groups A–C (i.e., Genotypes #1 and #4–5) of *O. sinensis* by Li et al. [41]. The possibility of producing circular RNA through backsplicing–exon circularization [73] should be considered during the study design because these distinct steric structures may have a considerable impact on reverse transcription PCR and cDNA sequencing. In addition, the ITS1-5.8S-ITS2 sequences of Genotypes #2 and #6 may adopt different types of steric conformations [4,37]. Thus, the design of other genotype-specific primers and the combined use of other molecular techniques should be considered.Wei et al. [21] reported the identification of a single teleomorph of AT-biased Genotype #4 of *O. sinensis* in both the fruiting body and mycelia of the caterpillar body of cultivated *C. sinensis* (unknown maturation stage) and a single teleomorph of GC-biased Genotype #1 in natural *C. sinensis*. The sequences of the two postulated teleomorphs of *O. sinensis* resided in distant phylogenetic clades (*cf.* Figure 6 of [21] and Figure 7 and Figure 8 of this paper). If AT-biased Genotype #4 represents a nonfunctional ITS pseudogene, as believed by Li et al. [41,42,58], the single AT-biased *O. sinensis* teleomorph in cultivated *C. sinensis* might have disturbed teleomorphic functions, leading to reproductive sterility and an abnormal, disturbed lifecycle of cultivated *C. sinensis*. Wei et al. [21] did not share any information about the formation of stromal fertile portion and ascospore production in cultivated *C. sinensis*, which is the critical feature of *O. sinensis* reproduction during the sexual life of natural and cultivated *C. sinensis* [8].Li et al. [4,5,28] reported the identification of teleomorphic Genotypes #4 and #15 of *O. sinensis* in AT-biased Cluster B (*cf.* Figure 8) in the stromal fertile portion densely covered with numerous ascocarps but not in ascospores collected from the same pieces of natural *C. sinensis* samples. AT-biased Genotypes #6 and #15 were found at high abundance in the stromal fertile portion prior to ascospore ejection, and their abundance drastically declined after ascospore ejection, whereas teleomorphic Genotype #5 maintained its high abundance in the stromal fertile portion before and after ascospore ejection [4,5,28].The 5.8S genes of multiple *O. sinensis* genotypes may be transcriptionally activated or silenced in different developmental and maturational stages of natural *C. sinensis*. Significant changes in the proteomic expression profiles of the stroma and caterpillar body of natural *C. sinensis* during maturation provide evidence of such dynamic alterations in the epigenetic, transcriptional, posttranscriptional, translational, and posttranslational modifications of numerous genes [74].

Because of the improper design of the study, Li et al. [41] provided insufficient and controversial evidence to support their “ITS pseudogene” hypothesis. The whole package of genes expressed in *O. sinensis* fungi and natural *C. sinensis* needs to be further explored before the 5.8S genes of AT-biased genotypes of *O. sinensis* can be defined as permanently nonfunctional pseudogenes.

## 4. Materials and Methods

### 4.1. Fungal Species in Natural C. sinensis

Information regarding intrinsic fungal species in natural *C. sinensis* was summarized by Jiang and Yao [8], and subsequent discoveries of additional species in natural *C. sinensis* were reviewed [4]. The sequences of 17 genotypes of *O. sinensis*, including the GC-biased Genotypes #1–3 and #7–14 and the AT-biased Genotypes #4–6 and #15–17, have been obtained and uploaded to GenBank by many research groups worldwide since 2001 and analyzed genetically and phylogenetically [4,5,16,18,19,20,21,22,23,24,25,26,27,30,37,38,39].

### 4.2. Genetic, Genomic, Transcriptomic, and Protein Sequences from GenBank

The GenBank database includes at least 668 *O. sinensis* ITS sequences under the GenBank taxid: 72228, 440 of which belong to GC-biased Genotypes #1–3 and #7–14 and 228 of which belong to AT-biased Genotypes #4–6 and #15–17 [4,5,19,23,25,28,30,39]. Pure cultures of *H. sinensis* (Genotype #1 of *O. sinensis*) were used for multigene, genome, transcriptome, and taxonomic analyses. However, pure cultures of Genotypes #2–17 of *O. sinensis* have not been obtained to date, which prevents genomic, transcriptomic, and multigene/multilocus analyses.

Five genome assemblies, ANOV00000000, JAAVMX000000000, LKHE00000000, LWBQ00000000, and NGJJ00000000, for the *H. sinensis* strains Co18, IOZ07, 1229, ZJB12195, and CC1406-203, respectively, are available in GenBank [44,46,47,48,49] and were used for the multilocus analysis of repetitive copies of authentic genes, including multiple GC- and AT-biased *O. sinensis* genotypes of the ITS locus.

The mRNA transcriptome assembly GCQL00000000 for the *H. sinensis* strain L0106 was uploaded to GenBank by Liu et al. [51]. Two metatranscriptome studies of natural *C. sinensis* have been reported. Xiang et al. [60] uploaded the GAGW00000000 metatranscriptome assembly to GenBank; this assembly was obtained from natural samples (unknown maturation stages) collected from Kangding County in Sichuan Province, China. Xia et al. [61] uploaded the assembled metatranscriptome sequences for mature *C. sinensis* samples collected from Deqin County in Yunnan Province, China, to www.plantkingdomgdb.com/Ophiocordyceps_sinensis/data/cds/Ophiocordyceps_sinensis_CDS.fas (accessed on 18 January 2018), which is currently inaccessible, but a previously downloaded cDNA file was used for analysis.

### 4.3. Annotations and Transcriptional Analysis of the Multilocus Authentic Genes in the Genome of H. sinensis in GenBank

GenBank does not provide annotations for authentic genes in the genome assemblies ANOV00000000, JAAVMX000000000, LKHE00000000, LWBQ00000000, and NGJJ00000000 of the *H. sinensis* strains Co18, IOZ07, 1229, ZJB12195, and CC1406-203, respectively [44,46,47,48,49]. One (JAACLJ010000002) of the thirteen genome contigs for *Ophiocordyceps camponoti-floridani* strain EC05 contained 1271 authentic genes [50], which was used to cross-reference, position, and annotate the authentic *H. sinensis* genes and their repetitive copies in the *H. sinensis* genome.

The transcription of the authentic *H. sinensis* genes and their repetitive genomic sequences, as well as the translated protein sequences, were compared with the mRNA transcriptome assembly GCQL00000000 for the *H. sinensis* strain L0106 [51] and the metatranscriptome assemblies for natural *C. sinensis* [60,61] using the GenBank BLAST algorithm (https://blast.ncbi.nlm.nih.gov/blast/ (frequently accessed)).

### 4.4. Genetic and Protein Sequence Analyses

We reanalyzed the multilocus DNA sequences and transcripts of authentic *H. sinensis* genes and their repetitive copies and the related protein sequences in the genome, transcriptome, and metatranscriptome assemblies of *H. sinensis* and natural *C. sinensis* via MegaBlast, discontinuous MegaBlast, Blastp, Tblastn, or blastn provided by GenBank (https://blast.ncbi.nlm.nih.gov/).

### 4.5. Phylogenetic Analyses

A Bayesian phylogenetic tree of the ITS sequences and their repetitive copies was inferred using MrBayes v3.2.7 software (Markov chain Monte Carlo [MCMC] algorithm) based on 3 × 10^4^ samples, with a sampling frequency of 10^3^ iterations after discarding the first 25% of the samples from a total of 4 × 10^6^ iterations [4,5,28,75]. This phylogenetic analysis was performed at Nanjing Genepioneer Biotechnologies Co.

### 4.6. Amino Acid Properties and Scale Analysis

The amino acid components of the proteins were scaled (Appendix A) and plotted sequentially at a window size of 21 amino acids for the α-helices, β-sheets, β-turns, and coils of the proteins using the linear weight variation model of the ExPASy ProtScale algorithm provided by the SIB Swiss Institute of Bioinformatics (https://web.expasy.org/protscale/) [43,52,53,54,55,56,57]. The overall amino acid hydrophobicity properties and the waveforms of the ProtScale plots for the proteins were compared to predict alterations in the secondary structures of the proteins.

## 5. Conclusions

The multilocus analysis revealed numerous repetitive copies of 104 of 1271 authentic *H. sinensis* genes. The repetitive sequences contained various transition and transversion alleles and insertion/deletion point mutations, causing decreases, increases, or bidirectional changes in the AT content. However, the repetitive copies were functionally transcribed, regardless of whether the point mutations were caused by RIP-related or non-RIP-related mutagenic evolutionary mechanisms. The repetitive ITS copies in the genome of Genotype #1 contained multiple scattered insertion, deletion, and transversion point mutations and may not be generated through RIP mutagenesis because, theoretically, RIP mutagenesis causes C-to-T and G-to-A transitions. The GC-biased repetitive ITS copies are genetically and phylogenetically distinct from AT-biased *O. sinensis* genotypes, which contain multiple scattered transition alleles. Thus, Genotype #1 *H. sinensis* may not be the target of RIP mutagenesis or concurrent epigenetic methylation attack. The AT-biased genotypes are independent of the genome of Genotype #1 and might have been generated from a common ancestor through certain mutagenic and evolutionary mechanisms during the long course of evolution in response to the extremely harsh environment of the Qinghai–Tibet Plateau. These genotypes became independent *O. sinensis* fungi with independent genomes in parallel with GC-biased genotypes in natural *C. sinensis*. Thus, Li et al. [41,42,58] provided unsound and controversial information to support their “ITS pseudogene” and the causative “RIP” mutagenesis hypotheses, which have not been validated by genomic and transcriptomic evidence.

## Figures and Tables

**Figure 1 ijms-25-11178-f001:**
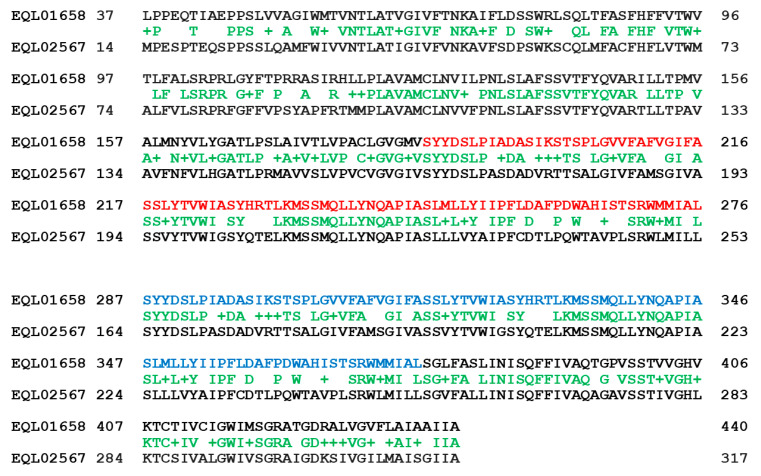
Alignments of the of the triose–phosphate transporter protein sequences EQL01658 and EQL02567 of *H. sinensis* strain Co18. The protein sequence EQL01658 (440 aa) encoded by the authentic gene of *H. sinensis* strain Co18 was aligned with the protein sequence EQL02567 (330 aa) encoded by the genomic repetitive sequence. The sequences in red and blue in both the upper and lower segments indicate the repeat segments within the EQL01658 sequence, which align to the same sequence segment of EQL02567. The letters and “+” symbols in green between the sequence lines refer to the identical and conservatively evolved amino acid residues when comparing the 2 protein sequences, respectively, and the spaces indicate non-conservatively variable amino acids when comparing the 2 protein sequences.

**Figure 2 ijms-25-11178-f002:**
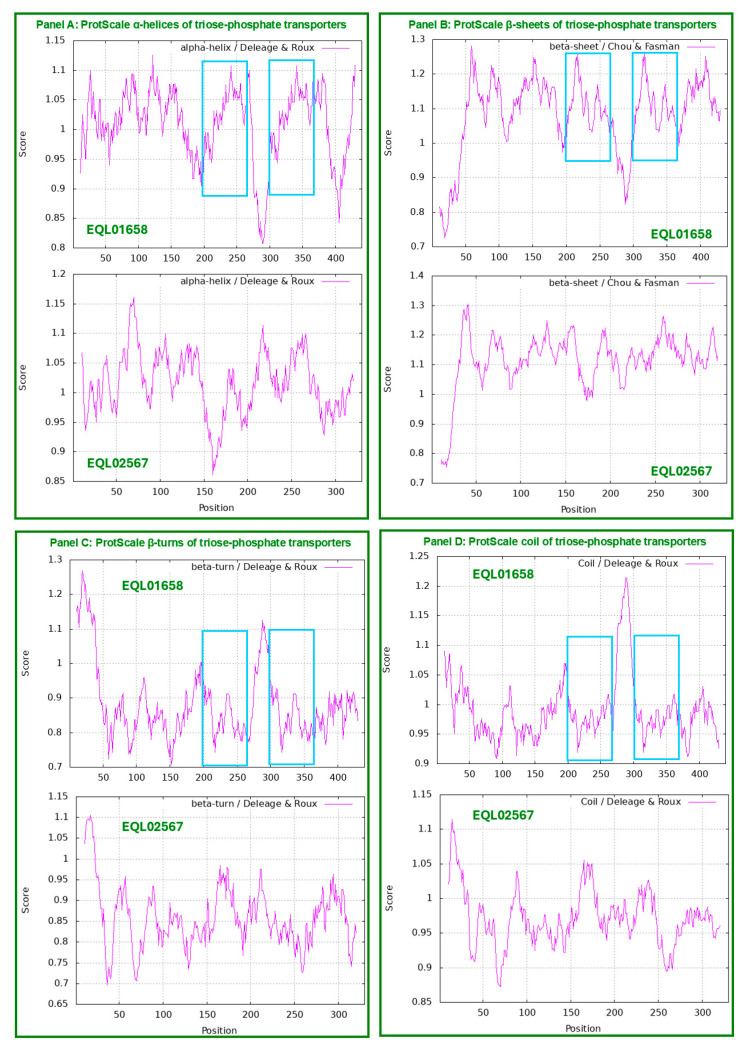
ExPASy ProtScale plots for α-helices (Panel **A**), β-sheets (Panel **B**), β-turns (Panel **C**) and coils (Panel **D**) of the triose-phosphate transporter protein. The protein sequence EQL01658 encoded by the authentic gene of *H. sinensis* strain L0106 was compared with the protein sequence EQL02567 encoded by the repetitive genomic sequence. The open boxes in blue in all the EQL01658 plots indicate the repeated protein sequence segments.

**Figure 3 ijms-25-11178-f003:**
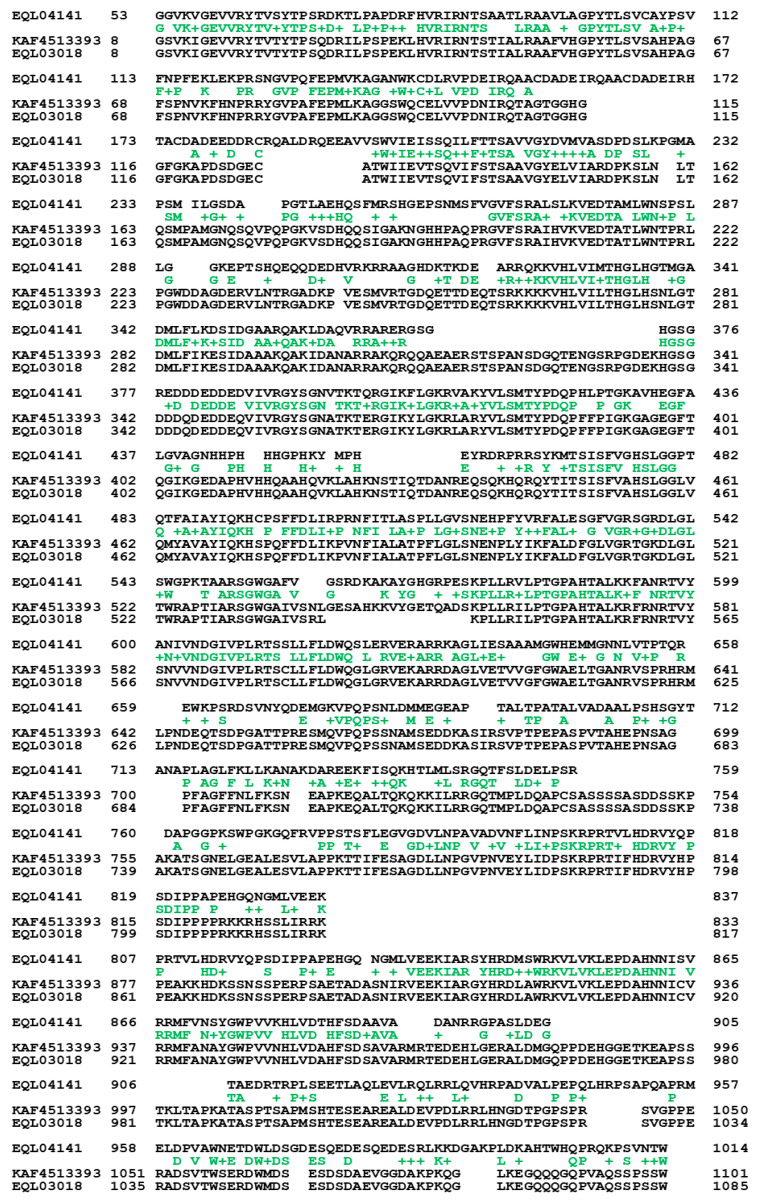
Alignments of the authentic lipase/serine esterase sequence EQL04141 of the *H. sinensis* strain Co18 with the KAF4513393 and EQL03018 sequences encoded by the repetitive genomic sequences. The protein sequence EQL04141 (1161 aa) encoded by the authentic gene for the lipase/serine esterase of *H. sinensis* strain Co18 was compared with the protein sequences KAF4513393 (1132 aa) and EQL03018 (1116 aa) encoded by the genomic repetitive sequences for other lipases or esterases. The letters and “+” symbols in green immediately above the sequence lines in black for KAF4513393 and EQL03018 refer to the identical and conservatively evolved amino acid residues when comparing the protein sequences encoded by the repetitive sequences, respectively, and the spaces indicate non-conservatively variable amino acids when comparing the protein sequences.

**Figure 4 ijms-25-11178-f004:**
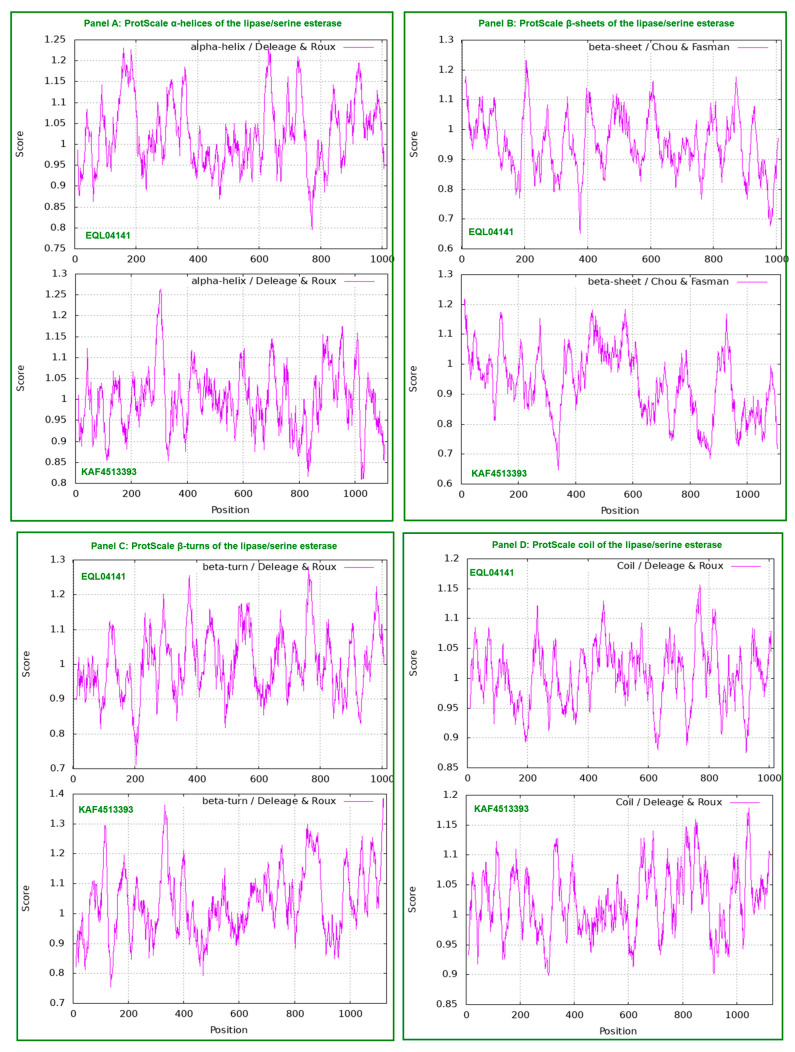
ExPASy ProtScale plots for α-helices (Panel **A**, containing 2 plots in pairs), β-sheets (Panel **B**), β-turns (Panel **C**), and coils (Panel **D**) of the esterase or lipase proteins. The authentic protein sequence EQL04141 encoded by the lipase/serine esterase gene of *H. sinensis* strain Co18 was compared with the protein sequence KAF4513393 encoded by a genomic repetitive sequence for other lipases or esterases.

**Figure 5 ijms-25-11178-f005:**
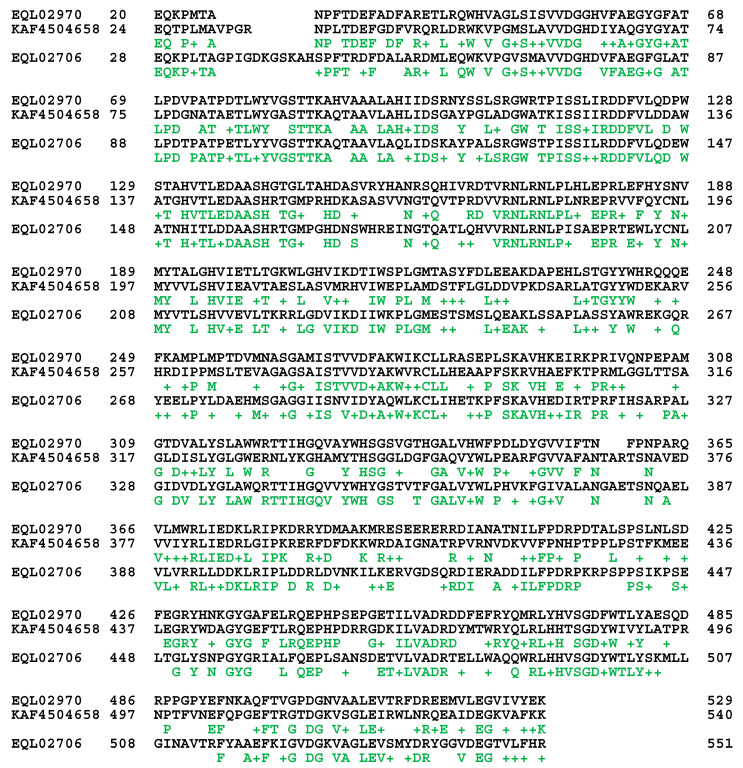
Alignments of the authentic β-lactamase/transpeptidase-like protein sequence EQL02970 of the *H. sinensis* strain Co18 with the KAF4504658 and EQL02706 sequences encoded by the repetitive genomic sequences. The protein sequence EQL02970 (531 aa) encoded by the authentic gene for the β-lactamase/transpeptidase-like protein of *H. sinensis* strain Co18 was compared with the protein sequences KAF4504658 (542 aa) and EQL02706 (558 aa) encoded by the repetitive genomic sequences. The letters and “+” symbols in green immediately below the repetitive sequence lines in black refer to the identical and conservatively evolved amino acid residues when comparing the protein sequences, respectively. The spaces in sequence lines in black stand for unmatched sequence gaps, and those in the green lines indicate non-conservatively variable amino acids when comparing the protein sequences.

**Figure 6 ijms-25-11178-f006:**
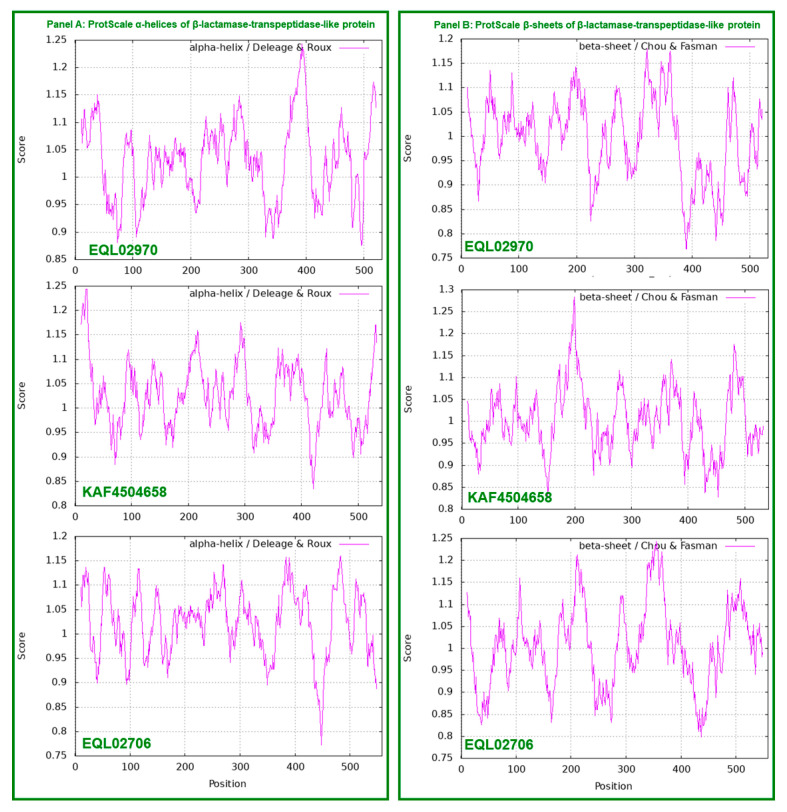
ExPASy ProtScale plots for α-helices (Panel **A**, containing 2 plots in pairs), β-sheets (Panel **B**), β-turns (Panel **C**), and coils (Panel **D**) of the β-lactamase/transpeptidase-like protein. The protein sequence EQL02970 encoded by the authentic gene encoding the β-lactamase/transpeptidase-like protein of *H. sinensis* strain Co18 was compared with the protein sequences KAF4504658 and EQL02706 encoded by the repetitive genomic sequences.

**Figure 7 ijms-25-11178-f007:**
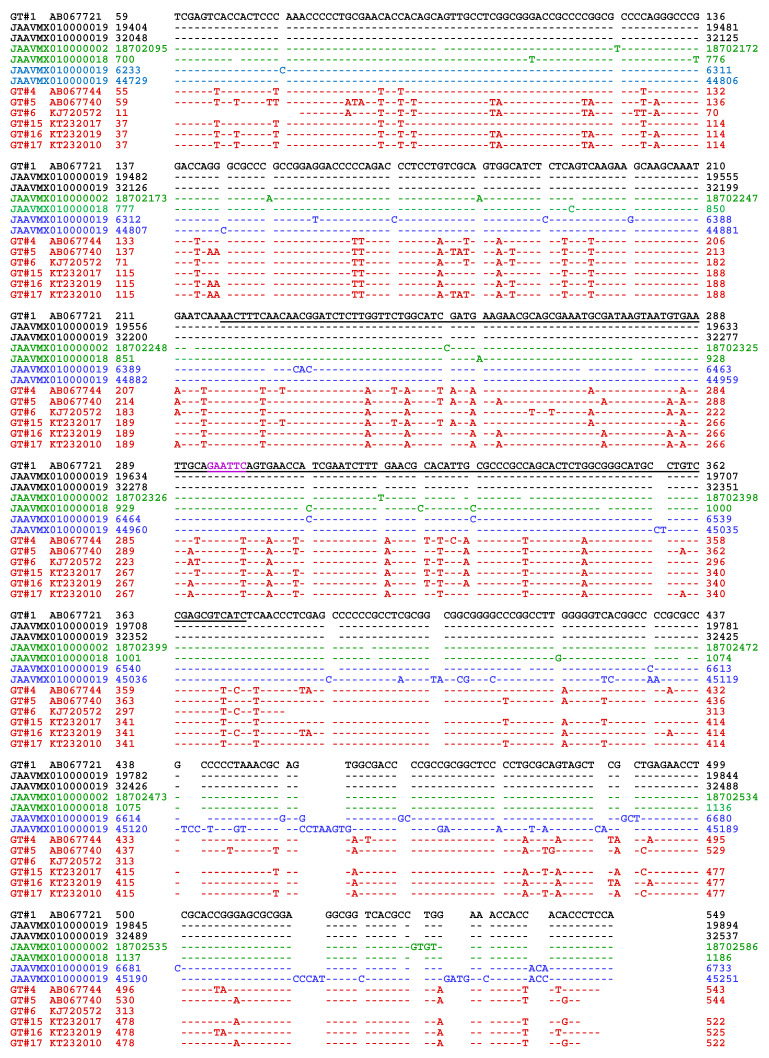
ITS sequence alignment of AB067721, variable and less variable repetitive ITS copies within the genome JAAVMX000000000 of the *H. sinensis* strain IOZ07, and AT-biased genotypes of *O. sinensis*. The ITS sequences contained complete or partial ITS1-5.8S-ITS2 nrDNA segments. “GT” denotes the genotype of *O. sinensis*. The underlined sequence in black represents the 5.8S gene of the GC-biased Genotype #1 of *H. sinensis*. AB067721 is the ITS sequence of GC-biased Genotype #1 of *H. sinensis*. The genome assemblies JAAVMX010000002, JAAVMX010000018, and JAAVMX010000019 were obtained from the *H. sinensis* strain IOZ07 [49]. One copy each within JAAVMX010000002 and JAAVMX010000018, indicated in green, shares 97.4% or 97.0% similarity with AB067721. JAAVMX010000019 contains 4 repetitive ITS copies, including 2 black sequences (19404→19894 and 32048→32537), which are 100% identical to AB067721, and 2 other blue sequences (6233→6733 and 44729→45251), with 94.5% and 90.8% similarity to AB067721. The sequences in red, namely, AB067744, AB067740, KJ720572, KT232017, KT232019, and KT232010, represent AT-biased Genotypes #4–6 and #15–17 of *O. sinensis*, respectively. The underlined “GAATTC” sequences in purple represent the EcoRI endonuclease cleavage sites in the sequences of GC-biased Genotype #1 and the GC-biased genome assembly JAAVMX000000000. EcoRI endonuclease cleavage sites are absent in AT-biased ITS sequences because of a single-base cytosine-to-thymine (C-to-T) transition. The hyphens indicate identical bases, and the spaces denote unmatched sequence gaps.

**Figure 8 ijms-25-11178-f008:**
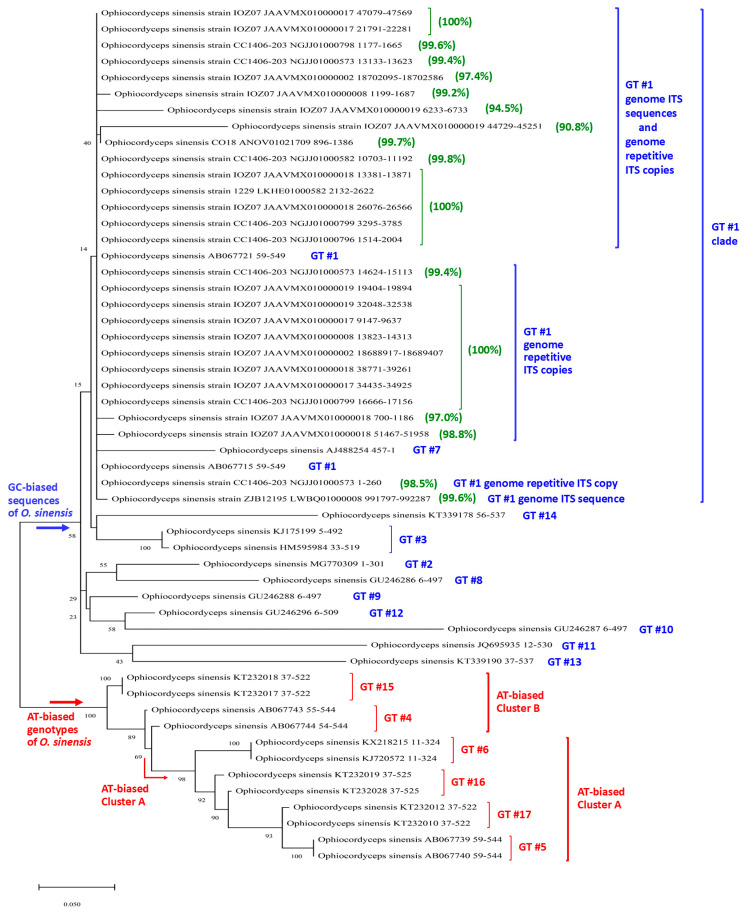
A Bayesian majority rule consensus phylogenetic tree. “GT” represents the genotype. Twenty-eight ITS segments within the genome assemblies (ANOV01021709, LKHE01000582, LWBQ01000008, JAAVMX010000002, JAAVMX010000008, JAAVMX0100000017, JAAVMX0100000018, JAAVMX010000019, NGJJ01000573, NGJJ01000582, NGJJ01000796, NGJJ01000798, and NGJJ01000799) of *H. sinensis* strains (Co18, 1229, ZJB12195, IOZ07, and CC1406-203, respectively)) and 25 ITS sequences of GC-biased Genotypes #1–3 and #7–14 (in blue alongside the tree) and AT-biased Genotypes #4–6 and #15–17 of *O. sinensis* (in red alongside the tree) were analyzed phylogenetically via MrBayes v3.2.7 software (*cf.* Section 2.4). Genome assemblies JAAVMX000000000 and NGJJ00000000 contain multiple repetitive ITS copies (*cf.* Table 5). The percent similarities of the genomic sequences of repetitive ITS copies with the representative Genotype #1 sequence (AB067721) are shown in green alongside the tree.

**Table 1 ijms-25-11178-t001:** Summary of 1271 authentic genes and their repetitive copies in the genomes of *H. sinensis* strains 1229, CC1406-203, Co18, IOZ07, and ZJB12195.

	# of Genes	Change in the AT Content		# of Genes Having Repetitive Copies
No genomic repetitive copies	1167 (91.8%)			
Single repetitive copy in only one of the 5 *H. sinensis* genomes	37 (2.9%)	Homologous (≥97% similarity)		36
		96.5% similarity	↓ 0.2%	1
Multiple repetitive copies in the genomes of the 5 *H. sinensis* strains	67 (5.3%)	Essentially no change	↑ or ↓ within ±1%	6
		Slight decreases	↓ ≤5%	11
		Large decreases	↓ >5%	7
		Slight increases	↑ ≤5%	17
		Large increases	↑ >5%	11
		Bidirectional changes		15
Total:	1271 (100%)			

Note: “**↑**” and “**↓**” indicate an increase or decrease in the AT content, respectively. The “#” symbols indicate the numbers of genes.

**Table 2 ijms-25-11178-t002:** Authentic *H. sinensis* genes encoding the triose-phosphate transporter (the query sequence) and repetitive genomic copies (the subject sequences) whose AT content decreased slightly (≤5%).

The Subject Sequence (Repetitive Copy)		*vs*. The Query Sequence (202613→203498 of NGJJ01001434 of the Authentic Gene for the Triose-Phosphate Transporter)		Mutation in the Subject Sequence Compared with the Query Sequence						Transcript in the mRNA Transcriptome GCQL00000000			
*H. sinensis* Strain	Genomic Fragment (Sequence Range)	Similarity	Change In AT Content	Transitions		Transversions		Total Point Mutations		Transcriptomic Fragment	Similarity	Query Coverage	Translated Protein Sequence
				C-to-T and G-to-A	T-to-C and A-to-G	C-to-A and G-to-T	A-to-C and T-to-G	G or C to A or T	A or T to G or C				
1229	LKHE01003487 (71694←72581)	99.9%	Nearly no change (↓ 44.2% to 44.1%)							GCQL01008460 (451→1271)	100%	92%	EQL01658 (67→439)
	LKHE01003657 (124108←124983)	68.1%	↓ 44.2% to 42.1%	62	70	21	58	83	128	GCQL01008629 (1→821) with a 48 nt deletion	94.3%	99%	EQL02567 (44→316)
CC1406-203	NGJJ01001434 (202613→203498)	100%	No change (44.2%)							GCQL01008460 (451→1080)	100%	92%	EQL01658 (67→376)
	NGJJ01000759 (751126→752000)	68.1%	↓ 44.2% to 42.1%	62	70	21	58	83	128	GCQL01008629 (1→821) with a 48 nt deletion	94.3%	99%	EQL02567 (44→316)
Co18	ANOV01001461 (5923→6447)	100%	No change (43.5%)							GCQL01008460 (810→1271)	100%	88%	EQL01658 (187→439)
	ANOV01001461 (5261→5919)	99.8%	Nearly no change (↓ 43.9% to 43.7%)							GCQL01008460 (451→1080)	100%	95%	EQL01658 (67→376)
	ANOV01000747 (27035→27910)	68.1%	↓ 44.2% to 42.1%	62	70	21	58	83	128	GCQL01008629 (1→821) with a 48 nt deletion	94.3%	99%	EQL02567 (44→316)
IOZ07	JAAVMX010000003 (1710624→1711511)	99.9%	Nearly no change (↓ 44.2% to 44.1%)							GCQL01008460 (451→1271)	100%	92%	EQL01658 (67→439)
	JAAVMX010000005 (9078524←9079399)	68.1%	↓ 44.2% to 42.1%	62	70	21	58	83	128	GCQL01008629 (1→821) with a 48 nt deletion	94.3%	99%	EQL02567 (44→316)
ZJB12195	LWBQ01000001 (1755599→1756257)	99.8%	Nearly no change (43.9%)	0	1	0	0	0	1	GCQL01008460 (451→1080)	100%	95%	EQL01658 (67→376)
	LWBQ01000001 (1756277→1756590)	98.4%	Nearly no change (↓ 43.5% to 43.3%)	2	1	1	0	3	1	GCQL01008460 (1022→1271)	100%	79%	EQL01658 (258→439)
	LWBQ01000030 (268003→268878)	68.1%	↓ 44.2% to 42.1%	62	70	21	58	83	128	GCQL01008629 (1←821) with a 48 nt deletion	94.5%	99%	EQL02567 (44→316)

Note: “→” and “←” indicate the sequence directions of the sense and antisense strands, respectively. “**↓**” indicates a decrease in the AT content.

**Table 4 ijms-25-11178-t004:** Authentic *H. sinensis* genes encoding the β-lactamase/transpeptidase-like protein (the query sequence) and repetitive genomic copies with bidirectional changes in the AT content.

The Subject Sequence (Repetitive Copy)		*vs*. The Query Sequence (70191→71142 of NGJJ01001580 of the Authentic Gene for the β-Lactamase/Transpeptidase-Like Protein)		Mutation in the Subject Sequence Compared with the Query Sequence						Transcript in the mRNA Transcriptome GCQL00000000			
*H. sinensis* Strain	Genomic Fragment (Sequence Range)	Similarity	Change in AT Content	Transitions		Transversions		Total point mutations		Transcriptomic Fragment	Similarity	Query Coverage	Translated Protein Sequence
				C-to-T and G-to-A	T-to-C and A-to-G	C-to-A and G-to-T	A-to-C and T-to-G	G or C to A or T	A or T to G or C				
1229	LKHE01000540 (19330→20932)	99.9%	No ∆ (38.7%)							GCQL01006885 (201←504) GCQL01011125 (1→411) GCQL01011658 (1152→1882) with a 61 nt deletion	100% 99.8% 92.1%	18% 25% 49%	EQL02970 (28→110) EQL02970 (140→275) EQL02970 (281→523)
	LKHE01001740 (54230←54311; 54418←55104)	67.9%	↓ 38.0% to 33.0%	68	68	26	40	94	108	GCQL01006426 (997→1188) GCQL01020269 (3→392) GCQL01008668 (979→1256)	100% 99.7% 98.2%	21% 45% 31%	KAF4504658 (45→70) KAF4504658 (57→184) KAF4504658 (208→297)
	LKHE01002381 (20238←21185)	64.2%	↑ 38.0% to 42.0%	74	46	48	43	122	89	GCQL01012138 (368→1264) with a 51 nt intron	94.6%	100%	EQL02706 (56→354)
IOZ07	JAAVMX010000002 (238583←240185)	100%	No ∆ (38.7%)							GCQL01006885 (201←504) GCQL01011125 (1→411) GCQL01011658 (1152→1882) with a 61 nt deletion	99.7% 99.8% 92.0%	18% 25% 49%	EQL02970 (28→110) EQL02970 (140→275) EQL02970 (281→523)
	JAAVMX010000009 (883241→883927; 884034→884115)	67.9%	↓ 38.0% to 33.0%	68	68	26	40	94	108	GCQL01006426 (997→1188) GCQL01020269 (3→392) GCQL01008668 (979→1256)	100% 99.7% 98.2%	21% 45% 31%	KAF4504658 (45→70) KAF4504658 (57→184) KAF4504658 (208→297)
	JAAVMX010000003 (16799894←16800841)	64.2%	↑ 38.0% to 42.0%	74	46	48	43	122	89	GCQL01012138 (368→1264) with a 51 nt intron	94.6%	100%	EQL02706 (56→354)
CC1406-203	NGJJ01001580 (70191→71142)	100%	No ∆ (38.7%)							GCQL01006885 (228←504) GCQL01011125 (1→411) GCQL01011658 (1718→1882)	99.6% 99.8% 99.4%	17% 43% 29%	EQL02970 (37→110) EQL02970 (140→275) EQL02970 (281→335)
	NGJJ01000083 (697372→698058; 698165→698246)	67.9%	↓ 38.0% to 33.0%	85	110	40	71	125	181	GCQL01006426 (997→1188) GCQL01020269 (3→392) GCQL01008668 (979→1256)	100% 99.7% 98.2%	21% 45% 31%	KAF4504658 (45→70) KAF4504658 (57→184) KAF4504658 (208→297)
	NGJJ01000732 (250601→251549)	64.2%	↑ 38.0% to 41.9%	74	46	48	43	122	89	GCQL01012138 (368→1264) with a 51 nt intron	94.5%	100%	EQL02706 (56→354)
Co18	ANOV01000528 (2857←4459)	99.8%	Nearly no ∆ (↑ 38.7% to 38.9%)	0	0	1	0	1	0	GCQL01006885 (201←504) GCQL01011125 (1→411) GCQL01011658 (1152→1882) with a 61 nt deletion	99.7% 99.8% 92.0%	18% 25% 49%	EQL02970 (28→110) EQL02970 (140→275) EQL02970 (281→523)
	ANOV01000033 (136→549; 656→737)	70.4%	↓ 35.7% to 32.7%	16	27	14	15	30	44	GCQL01006426 (997→1188) GCQL01020269 (3→392)	100% 99.7%	21% 65%	KAF4504658 (45→70) KAF4504658 (57→184)
	ANOV01000652 (15299→16246)	64.2%	↑ 38.0% to 41.9%	74	46	48	43	122	89	GCQL01012138 (368→1264) with a 51 nt intron	94.6%	100%	EQL02706 (56→354)
ZJB12195	LWBQ01000052 (168259←169259)	100%	No ∆ (40.2%)							GCQL01011125 (1←196) GCQL01011658 (1149→1882) with a 61 nt deletion	100% 92.2%	19% 79%	EQL02970 (212→275) EQL02970 (281→524)
	LWBQ01000002 (1407095←1407176; 1407283←1407696)	70.4%	↓ 35.4% to 33.0%	16	27	14	15	30	42	GCQL01006426 (997→1188) GCQL01020269 (3→392)	100% 99.7%	21% 65%	KAF4504658 (45→70) KAF4504658 (57→184)
	LWBQ01000004 (210989←211936)	64.2%	↑ 38.0% to 42.0%	74	46	48	43	122	89	GCQL01012138 (368→1264) with a 51 nt intron	94.6%	100%	EQL02706 (56→354)

Note: “→” and “←” indicate the sequence directions of the sense and antisense strands, respectively. “**↑**” and “**↓**” indicate increases and decreases in the AT content, respectively.

**Table 5 ijms-25-11178-t005:** GC content of the repetitive ITS1-5.8S-ITS2 sequences from the genome assemblies of 2 *H. sinensis* strains and percent similarities of the repetitive ITS sequences to the sequences of GC-biased Genotype #1 (AB067721) and AT-biased Genotypes #4–6 and #15–17 of *O. sinensis*.

Genome Segment	Sequence Range and Direction	GC Content	Percent Similarity of the Repetitive Sequence	
			vs. Genotype #1 AB067721	vs. AT-Biased Genotypes #4–6 and #15–17
JAAVMX010000002	18688917→18689407	64.8%	100%	86.7–89.9%
	18702095→18702586	64.5%	97.4%	82.8–88.0%
JAAVMX010000008	13823→14313	64.8%	100%	85.5–89.9%
	1199→1687	64.8%	99.2%	85.2–89.3%
JAAVMX010000017	9147←9637	64.8%	100%	85.5–89.9%
	21791←22281	64.8%	100%	85.5–89.9%
	34435←34925	64.8%	100%	85.5–89.9%
	47079←47569	64.8%	100%	85.5–89.9%
JAAVMX010000018	13381→13871	64.8%	100%	85.5–89.9%
	26076→26566	64.8%	100%	85.5–89.9%
	38771→39261	64.8%	100%	85.5–89.9%
	51467→51958	64.6%	98.8%	85.5–88.8%
	700→1186	64.5%	97.0%	82.6–87.6%
JAAVMX010000019	19404→19894	64.8%	100%	85.5–89.9%
	32048→32538	64.8%	100%	85.5–89.9%
	6233→6733	65.3%	94.5%	81.3–85.5%
	44729→45251	63.3%	90.8%	80.1–84.6%
NGJJ01000573	13133←13623	64.8%	99.4%	85.2–89.6%
	14624←15113	64.7%	99.4%	84.8–89.6%
NGJJ01000582	10703←11192	64.7%	99.8%	85.5–89.8%
NGJJ01000796	1514→2004	64.8%	100%	85.5–89.9%
NGJJ01000798	1177→1665	64.6%	99.6%	84.8–89.6%
NGJJ01000799	3295→3785	64.8%	100%	85.5–89.9%
	16666→17156	64.8%	100%	85.5–89.9%

Note: AB067721 represents the ITS1-5.8S-ITS2 sequence of GC-biased Genotype #1. The representative sequences for AT-biased genotypes are Genotype #4 AB067744, Genotype #5 AB067740, Genotype #6 KJ720572, Genotype #15 KT232017, Genotype #16 KT232019, and Genotype #17 KT232010. “→” and “←” indicate the sequence directions of the sense and antisense strands, respectively.

**Table 6 ijms-25-11178-t006:** Types of mutations in the variable and slightly less variable repetitive ITS sequences compared with those of GC-biased Genotype #1 of *H. sinensis* (AB067721).

	Sequence Range and Direction	% Similarity	Number of Mutant Alleles			Ins./Del./Transv. vs. Transit.
			Ins./Del.	Transv.	Transit.	
JAAVMX010000002	18702095→18702586	97.4%	13	0	0	13:0
JAAVMX010000018	700→1186	97.0%	10	5	0	15:0
JAAVMX010000019	6233→6733	94.5%	22	4	1	26:1
	44729→45251	90.8%	32	11	5	43:5

Note: “Ins./Del.” indicates insertion/deletion point mutations. “Transv.” denotes transversion point mutations. “Transit.” refers to transition point mutations.

## Data Availability

The analytical data is contained within the article and Appendix A. The original sequence data were from GenBank database.

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
