# Peer review of "Translation of Mutant Repetitive Genomic Sequences in *Hirsutella sinensis* and Changes in the Secondary Structures and Functional Specifications of the Encoded Proteins"

_ijms, 2024, doi:10.3390/ijms252011178_

Round 1

Reviewer 1 Report

Comments and Suggestions for Authors

Dear Editor of the journal Molecular Biology,

The manuscript titled" Translation of mutant repetitive genomic sequences in Hirsutella sinensis and changes in secondary structures and functional specifications of the encoded proteins" idea is quite good and the tools used in the sequence analysis are choosing in good manner. But I have some comments in this work if the authors would like to accept it.

First: Title is too long and should be reduced.

Abstract: is too long and contains a huge number of data, authors are not obligate to put all the obtained results in the abstract. They only can share to the end results and their role as a key in further investigations.

Keyword: is too much, should be reduced.

The aim of the work is not clear and should be listed.

Introduction: too long and should contains a paragraph dealing with transposons and their role in mutation in the different genomes.

Materials and methods: should contain a huge number of references and one recent reference is enough.

Table is S1: I see it can be listed in the results part.

Results: is quite good and it is written in good manner.

The alignment in page 8, I see it is not necessary and could be listed in supplementary data. The same for the figure FS5.

Discussion: written in very good manner but it needs more citation especially the recent ones. Also, author can reduced it because it is too long.

Conclusion: is too long and should be summarized.

Author Response

Responses to Reviewer #1:

1). You commented, “The manuscript titled "Translation of mutant repetitive genomic sequences in Hirsutella sinensis and changes in secondary structures and functional specifications of the encoded proteins" idea is quite good and the tools used in the sequence analysis are choosing in good manner. But I have some comments in this work if the authors would like to accept it.”

Thank you for the positive general review of our manuscript.

2). You commented, “First: Title is too long and should be reduced.”

We agree that the title is rather long. However, it was not a simple matter to shorten the title while maintaining the intended meaning.

The motivation for conducting this bioinformatic study was to prove or disprove the “ITS pseudogene” hypothesis for Hirsutella sinensis [Li et al. 2013, 2019] and the related “RIP” mutagenesis hypothesis [Li et al. 2020c] in the research of Ophiocordyceps sinensis. These authors suggested that the mutant AT-biased genotypes of O. sinensis were nonfunctional repetitive copies in the single genome of GC-biased Genotype #1 of O. sinensis generated through RIP mutagenesis. Please refer to the 2nd paragraph of the Introduction regarding the 17 mutant genotypes of O. sinensis. Thus, our study aimed to examine whether RIP-related C-to-T and G-to-A transitions occur in the repetitive genomic sequences of H. sinensis and whether concomitant epigenetic methylation disturbed the transcription and functional expression of the repetitive genomic sequences.

We have tried to shorten the title of our paper to “Translation of mutant repetitive genomic sequences in Hirsutella sinensis and changes in secondary structures and functional specifications of the encoded proteins”. However, removing any of the words in the title would alter or eliminate important meaning, and changing the syntax or phrase structure of the title might deviate from the main theme of our study: (1) transcription of repetitive genomic sequences and (2) functionality of the encoded proteins. Thus, we prefer not to alter the title further.

3). You commented, “Abstract: is too long and contains a huge number of data, authors are not obligate to put all the obtained results in the abstract. They only can share to the end results and their role as a key in further investigations.”

In accordance with your comment, we revised and shortened the Abstract.

4). You commented, “Keyword: is too much, should be reduced.”

We listed 5 key words, which are separated by “;” semicolons and should be considered reasonable number of keywords.

5). You commented, “The aim of the work is not clear and should be listed.”

We believe that you might have commented regarding the sentence in Lines 152-154 of the first submission. In accordance with your comment, we revised this passage to clarify the study aim:

“…the present bioinformatic study examined the repetitive sequences of 104 of 1271 authentic genes at multiple loci of the genome and transcriptome of GC-biased Genotype #1 H. sinensis and determined whether the H. sinensis genome is the target of RIP mutagenesis and concomitant epigenetic methylation, resulting in multiple repetitive sequences as genomic pseudogenes with loss of the transcription and translation functions”.

6). You commented, “Introduction: too long and should contains a paragraph dealing with transposons and their role in mutation in the different genomes.”

The Introduction of our paper is indeed quite long because of the complexity and controversy of C. sinensis studies. Unfortunately, the research progress in the field of natural C. sinensis has largely detached from the frontiers of the world's biological research community. Because of the low availability of raw material to scientists in the international community, mycological and Latin nomenclature studies by Chinese mycologists have generated many assumptions and hypotheses based on incomplete or even biased information, leading to scientific confusion. Owing to the enormous economic value of natural C. sinensis, this confusion has spread to the news media and the mass market, causing significant decreases in the wholesale and retail prices of natural products and relevant products. Thus, we need to retain as much detailed information as possible in the Introduction to clarify the existing confusion in the field to help our readers better understand our bioinformatic paper.

The first and second paragraphs (Lines 52-109 of the first submission) provide general information on the natural C. sinensis insect‒fungi complex and O. sinensis, which is actually not a single fungus. We have clarified the existing confusion, particularly the Latin name used in our paper. The Latin name O. sinensis represents 17 genotypes, the sequences of which exist in independent genomes belonging to multiple fungi. Among the 17 genotypes, Genotype #1 H. sinensis is the only one that has been purified and taxonomically characterized, whereas GC- and AT-biased Genotypes #2-17 of O. sinensis have not been purified or taxonomically characterized.

The third paragraph (Lines 110-129 of the first submission) describes the background information for the “ITS pseudogene” hypothesis under the molecular heterogeneity of O. sinensis and the multicellular heterokaryotic structure of natural C. sinensis. This hypothesis triggered a scientific debate because it was based on insufficient and controversial background information. This constitutes the fundamental reason for the current bioinformatic research.

The fourth and fifth paragraphs (Lines 130-149 of the first submission) describe the debate and recent research development of multiple O. sinensis genotypes. The “ITS pseudogene” hypothesis was initially developed but ultimately failed to explain the existence of multiple O. sinensis genotypes, particularly AT-biased O. sinensis genotypes, which constitute the controversial center of the scientific debate.

The sixth paragraph (Lines 150-160 of the first submission) outlines the main aim and general content of the current bioinformatic study.

Although the Introduction of our paper is rather long, we believe that the information provided is necessary for our readers to understand the bioinformatic research project and results presented in our paper. Removing any part of the Introduction will cause misunderstandings and confusion about the results of our paper.

6). You commented, “Materials and methods: should contain a huge number of references and one recent reference is enough.”

Yes, indeed. The Materials and Methods section did contain many references because of the nature of this bioinformatic study.

7). You commented, “Table is S1: I see it can be listed in the results part.”

Table S1 provides amino acid scaling information from the literature (α-helices, β-turns, coils [Deleage & Roux 1987] and β-sheets [Chou & Fasman 1978]), which was not generated from our own study. We used amino acid scaling information to compute sequential hydrophobicity to represent the secondary structures of proteins and to compare the topology and waveform alterations of the authentic proteins and their repetitive genomic copy counterparts, indicating potential alterations in the secondary structures and functional specificities of the proteins.

Because our paper contains so many tables and figures, we determined it would be better to put this table in the supplementary file.

8). You commented, “Results: is quite good and it is written in good manner.”

Thank you very much for your comment.

9). You commented, “The alignment in page 8, I see it is not necessary and could be listed in supplementary data. The same for the figure FS5.”

Do you mean Figure 7, ITS sequence alignment of AB067721, variable and less variable repetitive ITS copies within the genome JAAVMX000000000 of the H. sinensis strain IOZ07, and AT-biased genotypes of O. sinensis?

The AT-biased genotypes of O. sinensis are at the center of the scientific debate on the “ITS pseudogene” hypothesis and the consequences of RIP mutagenesis within a single genome of GC-biased Genotype #1 H. sinensis. The sequence alignment shown in Figure 7 is important in differentiating the point mutations occurring in AT-biased genotypic sequences and the GC-biased repetitive genomic sequences and in understanding that the AT-biased genotypes of O. sinensis are not the mutagenic products of RIP or pseudogenic components in the H. sinensis genome. Thus, we determined that Figure 7 should be retained in the main text and handled differently from Figure S5.

10). You commented, “Discussion: written in very good manner but it needs more citation especially the recent ones. Also, author can reduced it because it is too long.”

Thank you for the positive comment. We added 4 literature citations related to RIP mutagenesis and repetitive genomic sequences, which are publications from 2010, 2017, and 2022 and a Ph.D. candidate thesis from 2021. Accordingly, we revised the corresponding Discussion section.

11). You commented, “Conclusion: is too long and should be summarized.”

We revised and shortened the conclusion section.

Reviewer 2 Report

Comments and Suggestions for Authors

This manuscript explores the complex genomic and transcriptomic variations in Hirsutella sinensis through bioinformatics approaches. Specifically, it focuses on the repetitive sequences present in the fungal genome, their mutations induced by repeat-induced point mutation (RIP), and the consequent structural and functional alterations in the encoded proteins. The authors further discuss phylogenetic distinctions between different genotypes of Ophiocordyceps sinensis and analyze their secondary protein structures

Authors may want to provide a deeper analysis of the mutational signatures characteristic of RIP in the repetitive sequences of H. sinensis. This would further substantiate the claim that RIP is a driving force behind the sequence variations observed.

The discussion, while strong in terms of phylogenetic and genomic specifics, misses broader implications of the findings. The study does not fully explore how the genomic variations in H. sinensis could impact therapeutic applications of C. sinensis or contribute to the fungi’s ecological adaptability. A broader discussion would help place these findings into a wider biological and medicinal context.

Author Response

Responses to Reviewer #2:

1). You commented, “This manuscript explores the complex genomic and transcriptomic variations in Hirsutella sinensis through bioinformatics approaches. Specifically, it focuses on the repetitive sequences present in the fungal genome, their mutations induced by repeat-induced point mutation (RIP), and the consequent structural and functional alterations in the encoded proteins. The authors further discuss phylogenetic distinctions between different genotypes of Ophiocordyceps sinensis and analyze their secondary protein structures.

Authors’ Response: Thank you for the positive comment on our manuscript.

2). You commented, “Authors may want to provide a deeper analysis of the mutational signatures characteristic of RIP in the repetitive sequences of H. sinensis. This would further substantiate the claim that RIP is a driving force behind the sequence variations observed.”

Authors’ Response: Unfortunately, the results of our multilocus analysis indicate that O. sinensis is not the target of RIP mutagenesis or concomitant epigenetic methylation attacks. The repetitive genomic copies of authentic genes in the H. sinensis genome are normally transcribed, and their encoded proteins have minor alterations in secondary structures and function specifications. Thus, the repetitive sequences are the products of a long course of evolution but not of RIP mutagenesis immediately before or after a new H. sinensis haploid is generated.

3). You commented, “The discussion, while strong in terms of phylogenetic and genomic specifics, misses broader implications of the findings. The study does not fully explore how the genomic variations in H. sinensis could impact therapeutic applications of C. sinensis or contribute to the fungi’s ecological adaptability. A broader discussion would help place these findings into a wider biological and medicinal context.”

Authors’ Response: We have revised our manuscript in accordance with your comment.